# Study on the Origin and Classification of Two Poplar Species on the Qinghai–Tibet Plateau

Yu-Jie Shi, Jia-Xuan Mi, Jin-Liang Huang, Fang He, Liang-Hua Chen and Xue-Qin Wan *

Sichuan Province Key Laboratory of Ecological Forestry Engineering on the Upper Reaches of the Yangtze River, College of Forestry, Sichuan Agricultural University, Chengdu 611130, China; 2019104006@stu.sicau.edu.cn (Y.-J.S.); s20177704@stu.sicau.edu.cn (J.-X.M.); huangjinliang8886@gmail.com (J.-L.H.); 14686@sicau.edu.cn (F.H.); sicauchenlh@126.com (L.-H.C.)
* Correspondence: w-xue@163.com

**Abstract:** Poplar not only has important ecological and economic value, but also is a model woody plant in scientific research. However, due to the rich morphological variation and extensive inter-specific hybridization, the taxonomy of the genus *Populus* is very confused, especially in the Sect. *Tacamahaca.* Based on the extensive investigation of *Populus* on the Plateau and its surrounding areas, two taxa (*Populus kangdingensis* and *Populus schneideri* var. *tibetica*) that are very similar in morphology and habitat and are in doubt in taxonomy were found. First of all, we set up 14 sample sites, carried out morphological investigation and statistics, and found that there were a few morphological traits that could be distinguished between the two taxa. Further phylogenetic analysis based on the whole genome resequencing data showed that the two taxa were hybrid progenies of *P. xiangchengensis* and *P. simonii*. Through gene flow detection and genetic differentiation analysis, it was found that there was still strong gene flow from *P. xiangchengensis* to the two taxa, and there was almost no differentiation between the two taxa. Therefore, *P. schneideri* var. *tibetica* should be classified into *P. kangdingensis* as same taxa. Finally, the population history was reconstructed by PSMC and ABC models, and it was found that they all belonged to a hybrid origin, and the change in population size was closely related to the Quaternary ice age. In addition, the hybrid population has better adaptability, and the suitable distribution area may expand in the future. This study provided a novel and comprehensive method for the phylogeny of *Populus* and laid a foundation for the development and utilization of poplar resources.

**Keywords:** classification; *Populus*; morphology; genomics; origin; adaptability

## 1. Introduction

Poplar are various important woody plants of the genus *Populus* in Salicaceae. All the species in the genus are deciduous trees, which are widely distributed in the forests from the subtropics to the north of the Northern Hemisphere, mainly in temperate forests [1,2]. With the characteristics of fast growth, easy reproduction, and strong adaptability, it is one of the most important afforestation tree species in the world, and plays an important role in global wood production and ecological environment construction [1,3]. In addition, poplars also have good experimental characteristics, such as dioecious plants, easy hybridization, high compatibility, rapid growth, asexual (vegetative) reproduction along with the sexual one, and a small genome (about 450 Mb), which makes them ideal objects for genetic breeding research [4–7]. In the past few decades, the taxonomy, phylogeny, and origin of *Populus* have been widely studied by scholars at home and abroad. In the internationally accepted North American classification system, the genus *Populus* is divided into six sections and about 29 species [2], while the classification system in China divides it into five sections and about 71 species [8]. Due to the great difference between the Chinese classification system and the North American classification system, there is a great controversy among different scholars on the classification of the genus *Populus*. Moreover, even within China, there is a

great controversy among different scholars on the classification of *Populus*. These conditions are all very disadvantageous to poplar breeding, cultivation, and scientific research.

The West Sichuan Plateau is located in western Sichuan and southeast of the Qinghai–Tibet Plateau (QTP). Because of its complex and diverse geographical environment, diverse climatic conditions, and unique geological and historical conditions, especially in the Hengduan Mountains when the ice age was approaching in the fourth century, it not only provides excellent conditions for the retreat of plants [9,10] but also provides a channel for the retreat of poplars, so that the region is rich in poplar resources [11]. Among them, the poplar of Sect. *Tacamahaca* is absolutely dominant, and is one of the centers of natural distribution and evolution of poplars in China [12]. *Populus kangdingensis* and *Populus schneideri* var. *tibetica* are two taxa of Sect. *Tacamahaca*, which are mainly distributed on the surface of Plateau from 3400 to 4000 m above sea level in western Sichuan. They are also native poplar species endemic to southwest China, which are widely planted and play an important role in Plateau ecological construction, landscape construction, and production of medium and small-diameter wood and paid carbon wood [11,13]. *P. kangdingensis* was identified and named by Wang, et al., in 1979 [14], and *P. schneideri* and *P. schneideri* var. *tibetica* were named by Chao in 1985. Flora of China, edited in 1999, contains *P. schneideri*, but not *P. schneideri* var. *tibetica* [8]. In terms of distribution, Flora of Sichuan describes the distribution of *P. schneideri* var. *tibetica* in Yunnan and Tibet, but there are no records in the Flora of Yunnan and Flora of Tibet. In the poplar academia and forestry industry, it is generally believed that *P. schneideri* var. *tibetica* are only distributed in the narrow areas of the western Sichuan Plateau, but not outside Sichuan. Therefore, the classification and distribution of *P. schneideri* var. *tibetica* are greatly controversial.

The *P. kangdingensis* and *P. schneideri* var. *tibetica* have excellent adaptability to alpine and arid habitats, so they have important breeding value in poplar breeding facing QTP. However, the development, utilization, and scientific research on these taxa require an accurate classification. At present, there is no special research on them, although there are some studies on the classification, phylogeny, and breeding of *Populus tomentosa* [11,15]. Therefore, under the background of extensive investigation on the genus *Populus* (22 taxa) on the QTP, the habitat and morphological data of 70 individuals from 14 sample sites of *P. kangdingensis* and *P. schneideri* var. *tibetica* were collected, and 10 individuals of them were randomly selected for whole genome resequencing to further reveal their classification and origin. It lays a good foundation for the utilization and breeding of poplar resources.

## 2. Materials and Methods

### 2.1. Sample Sites Investigation and Material Collection

Based on the Flora of China and the local Flora of Sichuan, Yunnan, Tibet, Shaanxi, and Gansu, an extensive investigation was conducted on *Populus* plants in the Plateau of western Sichuan from 2006 to 2015. Since 2016, *Populus* has been investigated and collected many times in western Sichuan, northwestern Yunnan, southeastern Tibet, Shaanxi, and southern Gansu, which basically covers the natural distribution area of *Populus* in the QTP and its surrounding areas (Figure S1).

In the investigation, the genus *Populus* with population distribution was selected as the sample site for taxonomic investigation, including species identification, collection of specimens, infructescences, capsules, short branches with leaves, and related information records. The identification of all taxa is mainly based on the description of Flora of China, and local Flora. *P. kangdingensis* (*Ka*) and *P. schneideri* var. *tibetica* (*St*) are two taxa from the 22 taxa of *Populus* in the investigation. Based on considerable research, four sample areas were set up on the western Plateau of Sichuan, which basically covered the known distribution range of *Ka* and *St* (Figure 1). In this interval, above every 1 km, the poplar whose population is the distribution unit is selected as the survey sample site. A total of 14 sample sites were set up, including five *Ka* and nine *St*. After the sample site was set up, the basic information such as origin, geographical location, altitude, community composition, and habitat characteristics were recorded (Table S1).

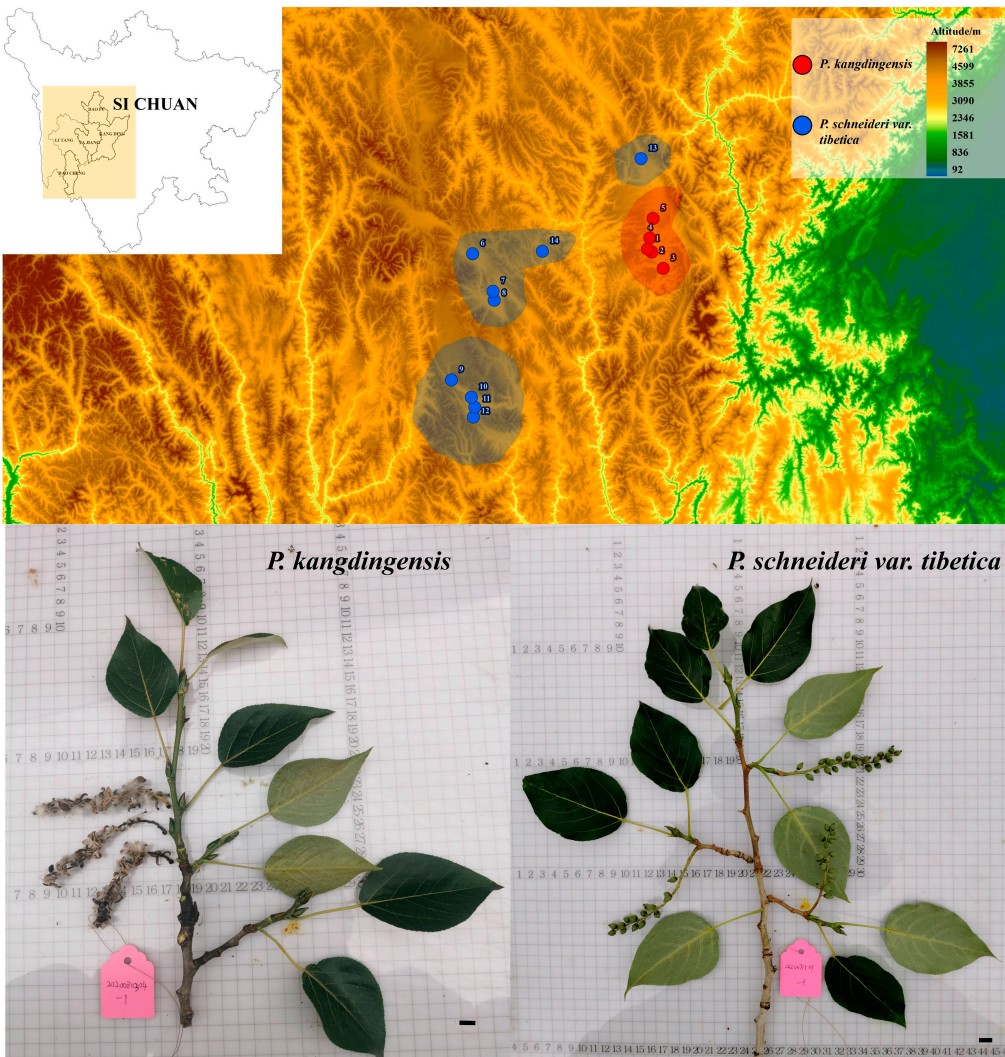

**Figure 1.** Sample sites distribution map, and specimen of *P. kangdingensis* and *P. schneideri* var. *tibetica*. Black bar = 1 cm.

In each sample site, at least 5 representative adult healthy female plants were selected as samples, and the infructescences and short branches with leaves were collected to determine the classification characteristics. More than 20 short branches with leaves under the canopy of the sample trees were selected to ensure that more than 50 typical intact leaves could be obtained from each tree (14 populations × 5 trees × 50 leaves). Due to the differences in altitude, latitude, and habitat conditions of the sample sites, the capsules of trees at most sample sites cracked and the infructescences had fallen to the ground at the time of the investigation, but in some cases, neither of these events had occurred. Therefore, for the plants whose infructescences were still hanging on the tree, 50 infructescences were randomly collected, while for the plants whose infructescences had fallen off, at least 50 intact and clean infructescences were picked up under the crown (14 populations × 5 trees × 50 infructescences). All samples of leaves and infructescences were collected and numbered, then packed in the sample bag and brought back to the room to determine the relevant traits. The corresponding cuttings of all samples were collected and brought back to the Modern Agricultural Research and Development Base of Sichuan Agricultural University (103°38′43″ E, 30°33′24″ N) for subsequent common garden experiments.

## 2.2. Determination of Morphological Traits

Ten quantitative morphological traits of easy-to-obtain and taxonomic significance were measured, including 5 traits related to leaves (petiole length, leaf length, leaf width, ratio of petiole length to total leaf length (the sum of leaf length and petiole length), and ratio of leaf length to leaf width), and 5 traits related to capsules (percentage of capsule valve cleavage, infructescences length, valve length, valve width, and ratio of valve length to valve width).

For leaf samples of each tree, 15 leaves were randomly selected from 50 intact leaves to measure petiole length, leaf length (excluding petiole), leaf width, and total leaf length (including petiole) with a ruler. Then, according to the above traits, the ratio of leaf length to leaf width and the ratio of petiole length to total leaf length were calculated. For capsule samples of each tree, 15 intact infructescences were randomly selected from 50 infructescences to measure the length of the infructescences with a ruler, then all the capsules from the aforementioned 15 infructescences were removed and placed in an oven at 40 °C for 24 h. After the capsules were completely cracked, at least 50 capsules were randomly selected. The length and width of the capsule valvular were measured with a Vernier caliper, and the ratio of valve length to width was calculated. Then, at least 100 samples were randomly selected from the completely cracked capsules, the percentage of the valve was counted one by one, and the proportion of 2-valve, 3-valve and 4-valve was calculated.

## 2.3. DNA Extraction and Resequencing

Because there are more polyphenol polysaccharides in the old leaves of poplar, the 106 individuals of 22 taxa (Table S2, including 5 individuals of *Ka* and 5 individuals of *St* in this study) were collected from the common garden, and the new young leaves of cuttings were collected for the extraction of DNA. After the young leaves were collected, they were quickly frozen in liquid nitrogen, and the whole genomic DNA of each plant was extracted by a slightly modified CTAB method (1 mL mixture of $V_{phenol}$:$V_{chloroform}$:$V_{isoamylol}$ = 25:24:1) [16]. Next, the 106 samples' DNA were sent to Novogene Co., Ltd. (Beijing, China) for genome resequencing. After passing the quality test of DNA, the fragments with a length of 350 bp were randomly broken by a Covaris crusher, and libraries were created using NEB Next® Ultra™ II DNA Library Prep Kit. Then, the constructed libraries were sequenced by the Illumina Novaseq™ platform with an average sequencing depth of 10× (Table S3).

The low-quality reads of raw reads were inspected and filtered using FastQC[9] and Trimmomatic v.0.40 [17]. All clean reads of each sample were mapped to the *P. trichocarpa* reference genome [6] using the BWA-MEM algorithm of BWA v.0.7.17 with default parameters [18]. The Sequence Alignment Map (SAM) format files were converted to Binary Alignment Map (BAM) format files using SAMtools v.0.1.19 [19]. Moreover, polymerase chain reaction duplicates were removed using Picard tools v.2.1.1. The Genome Analysis Toolkit (GATK) v.4.2.6.1 was used to call single nucleotide polymorphisms (SNPs) and short InDels with default settings from each species separately [20], and to join SNPs from all individuals. SNP sites were filtered according to mass value, site depth, Fisher test value, and comparison quality using GATK with the parameters as "QUAL < 30 || QD < 3.0 || FS > 60.0 || MQ < 40.0," and missing data were filtered using SAMtools with the parameters as "Dp3-miss0.2-maf 0.05" [19]. The final SNPs were annotated referring to the *P. trichocarpa* genome using ANNOVAR [21].

The individuals mapping to *P. trichocarpa* were prepared for two SNPs data sets: (1) a 22-taxa SNPs data set, which was used for phylogenetic TREE and STRUCTURE analysis; and (2) a 5-taxa SNPs data set including *Ka*, *St*, *P. simonii* (*Si*), *P. xiangchengensis* (*Xi*), and *P. haoana* (*Ha*), which was used for phylogenetic TREE, PCA, STRUCTURE, and subsequent hybridization, gene flow, and population history dynamic analysis.

### 2.4. Phylogenetic Relationship Analysis

Based on the neighbor joining (NJ) method, the whole genome phylogenetic trees of 22 poplar taxa were constructed by using MEGA-X software [22]. The bootstrap values were obtained after 1000 replications to test the accuracy of the phylogenetic tree. Finally, the phylogenetic tree was visualized by Chiplot online tool (https://www.chiplot.online/, accessed on 14 March 2023). Then, the genetic structure was analyzed by Admixture [23] software ($K$ = 2–20), and the optimal $K$ value was estimated by cross-validation ($CV$) error and visualized by TBtools [24]. Based on the above analysis, the taxa related to the phylogeny of *Ka* and *St* were selected for phylogenetic tree construction, principal component analysis (PCA), and genetic structure analysis. First of all, a maximum likelihood (ML) phylogenetic tree of whole-genome SNPs was constructed using IQ-TREE v2.2.0.3 [25]. Next, ModelFinder was used to determine the optimal substitution model and corresponding parameters according to three rules (AIC, AICc, and BIC) in IQ-TREE [26]. Then, the phylogenetic tree was inferred according to 1000 ultrafast bootstrap approximations [27]. Finally, the phylogenetic tree was visualized using the Chiplot. The method of genetic structure analysis is the same as above, except that the $K$ value is from 2 to 5. PCA was performed using Eigensoft Smartpca [28] based on the whole-genome SNPs dataset with default parameters; this was visualized using R packages.

### 2.5. Gene Flow and Hybridization Detection

ABBA-BABA analysis was performed to test for evidence of introgression of *Populus* using the Dsuite v.0.5 program [23], and Z-scores > 3 were considered statistically significant. This analysis used ancestral and derived allelic patterns in both ingroups and outgroups to distinguish between incomplete lineage sorting and hybridization. In short, for the ordered [((*P1*, *P2*), *P3*), *O*], the ABBA locus pattern refers to the shared derived alleles of *P2* and *P3*, and the BABA locus pattern refers to the shared derived alleles of *P1* and *P3*. The *D*-statistics were calculated as (ABBA − BABA)/(ABBA + BABA). Under the null hypothesis of incomplete lineage sorting (ILS), the numbers of ABBA and BABA were expected to be equal ($D$ = 0). Alternatively, a significant deviation of $D$ from zero would indicate that other events had occurred, particularly *P3* swapping genes with *P1* or *P2* [29,30]. According to previously described protocol [31], we first performed the ABBA-BABA detection of the four-taxon network from four groups [(*Xi*, *St*, *Si*, *O*); (*Xi*, *Ka*, *Si*, *O*)], then fixed the position of *F2* population, exchanged the population of *F1* and *F3* [(*Si*, *St*, *Xi*, *O*); (*Si*, *Ka*, *Xi*, *O*)], and then carried out ABBA-BABA test again. If the $D$ value of the two tests is greater than 0 and significant, it shows that *F2* may be the hybrid offspring of *F1* and *F3*.

We also used HyDe [32], a software package for detecting hybridization using phylogenetic invariants arising under the coalescent model with hybridization. Similar to ABBA-BABA detection, HyDe considers a rooted four-taxon network consisting of an outgroup and a triple ingroup population. Briefly, for the ordered model [((*P1*, Hybrid), *P2*), *O*], the gene trees arise within the parental population trees following the coalescent process, where the hybrid population is either sister to *P1* with a probability, 1-γ, or sister to *P2* with a probability, γ. We performed HyDe tests for two *Ka* and *St*, with *P. wilsonii* (*Wi*) as an outgroup. Using three species-level hypothesis tests, we assumed these species to be hybrid species. All results from the HyDe tests were filtered using 1% critical value Z-scores. Finally, we used TreeMix v.1.13 software [33] to infer the population splitting and mixing of four poplar species, including *Xi*, *Si*, *Ka, and St*, using *Wi* as an outgroup.

### 2.6. Population Genomic Analyses in Sliding Windows

To calculate the nucleotide diversity within populations ($p_i$), the absolute divergence coefficient ($D_{xy}$) and relative differentiation coefficient ($F_{st}$) were calculated using whole genomic data from each species using the pixy [34] in 100 kb non-overlapping sliding windows. Moreover, for pairwise species, Tajima's $D$ statistics over the 100 kb non-overlapping window were calculated using VCFtools v.0.1.13 [35].

We also performed a pairwise identity-by-descent (IBD) block analysis based on whole genome SNPs using Beagle v.4.1 [36] with the following parameters: window = 50,000; overlap = 5000; ibdlod = 5; ibdtrim = 100.

### 2.7. Population Historical Dynamics Analysis

Firstly, we used the PSMC model to infer the historical dynamics of $N_e$ with parameters -N25 -t15 -r5 -p "4 + 25 × 2 + 4 + 6" for each individual [37]. Then, assuming a generation time of 15 years and a mutation rate of $3.75 \times 10^{-8}$ site/year [38] for each species, we selected all individuals to perform the PSMC analyses and used R scripts to visualize the graphics.

Next, we constructed and compared three hypothetical species formation models through the analysis of the whole genomic SNP data using DIYABC Random Forest (DIYABC-RF) v.1.2.1 [39] for *Ka* and *St* groups [pop1, pop3, pop2]: 1) [*Si*, *Ka*, *Xi*]; 2) [*Si*, *St*, *Xi*]. In all three models (Figure S9 and Table S5), $T_A$ represented the divergence time between ancestral populations of pop1 and pop2, and $N_A$ represented the effective population size of the common ancestor of those species. Model 1 (hybrid origin) assumed a hybrid event between pop1 ($r_a$) and pop2 ($1 - r_a$), giving birth to pop3 at $T_1$. Model 2 (diverged from pop1), $T_2$ indicated the diverging time of pop3 and pop1; Model 3 was just the opposite of Model 2. Model 3 (diverged from pop2), $T_2$ indicated the diverging time of pop3 and pop2.

The SNP data set needs to be processed before the ABC model test. First, we filter the SNP sites according to linkage disequilibrium (LD) to retain the unlinked SNP sites. Moreover, we used Beagle v.4.1 software to fill in the missing data [40] and then used Python script vcf2diyabc.py to convert the VCF format into DIYABC-RF software-specific format. Finally, we obtained 65438 and 62666 SNPs with MAF > 0.05 criterion from three groups of SNP data sets [(*Si*, *Ka*, *Xi*); (*Si*, *St*, *Xi*)], respectively. We performed 120,000 simulated datasets (i.e., approximately 40,000 per scenario) for three groups of SNP data sets, respectively. The number of trees in the constructed Random Forest was fixed to 3000 according to the USER MANUAL for DIYABC-RF, as this number turned out to be large enough to ensure a stable estimation of the global error rate (Figure S8). We predicted the best scenario and estimated its posterior probability, as well as the global and local error rates of RF analyses based on the same training set. Then, we estimated each parameter of the best-fitting scenario (i.e., the selected scenario after processing scenario choice with DIYABC-RF) using 40,000 simulation data. The number of trees in the constructed Random Forest was fixed at 3000. For each parameter, we conducted five replicate RF analyses based on the same training set (Table S5).

### 2.8. Positive Selection Analysis

First of all, the top 5% values were screened out by using the $F_{st}$ between hybrids and two parents. Then the intersection of the above two $F_{st}$ was selected to indicate that there were highly differentiated gene fragments between the hybrid and the two parents. Finally, the most homologous genes in *Populus* and *Arabidopsis thaliana* were found by protein BLAST, and the gene function annotation and enrichment analysis were carried out based on GO and KEGG databases.

### 2.9. Prediction of Suitable Distribution Area of Species

Firstly, 19 global climate factors of four periods (the Last Interglacial, the Last Glacial Maximum, the Current, and 2061–2080) are downloaded from the WorldClim database (https://www.worldclim.org/, accessed on 24 April 2023), and their spatial resolutions are all 2.5 m. In order to avoid the over-fitting of the model caused by multiple collinearities among environmental factors, according to the contribution rate of model training and the results of Pearson correlation analysis from environmental factors in SPSS, we eliminate the environmental factors whose correlation coefficient is higher than 0.8, and contribution rate is low. Then, combined with the coordinate information of the sample points of poplar, the fitting analysis of the suitable distribution area was carried out using MaxEnt v.3.4.3 [41].

The 25% distribution data is selected as the detection data of the model; 75% data is used as training data, repeated 10 times, and the other parameters are set as default values for modeling. Finally, based on the fitted mean value, ArcGIS v.10.8 is used to visualize the suitable distribution area.

*2.10. Statistical Analysis of Data*

The morphological traits of leaves and capsules were described and analyzed by IBM SPSS v26. The Student's *t*-test was used to compare the interspecific data, and the significance was expressed by * ($p < 0.05$) and ** ($p < 0.01$). The data of more than two groups were analyzed by the Duncan algorithm of one-way ANOVA, and the significance was expressed by different lowercase letters. Canonical discriminant analysis was also performed through the SPSS software. In addition, the systematic clustering analysis based on morphological indicators was performed by Origin software, in which the clustering method is set as the Average method, the distance type is set as Pearson correlation, and the other parameters are default.

## 3. Results

*3.1. Distribution and Habitat of Two Taxa of Poplar*

After our full investigation of the *Populus* in the QTP, it is found that the actual distribution ranges of the two taxa of poplar are very narrow. *Ka* is mainly distributed in the Kangding–Daofu–Luhuo–Ganzi line of the western Sichuan Plateau, at the bottom valley from 3400 to 3600 m, especially in Xinduqiao Town. *St* is also mainly distributed on the Plateau surface from 3600 to 4000 m in the western Sichuan Plateau, except for sporadic distribution in Yajiang County and Daofu County. It is concentrated along Daocheng County to Litang County. There is no distribution of *Ka* and *St* in other areas except Sichuan Province. Therefore, a total of 14 sample sites were set up, all of which were located in the Ganzi Prefecture of West Sichuan Plateau (Figure 1 and Table S1).

*3.2. Morphological Traits of Two Taxa of Poplar*

In the leaf morphological traits of *Ka* and *St*, the leaf length range is from 7.28 to 9.88 cm and 7.97 to 9.64 cm, the leaf width range is from 4.81 to 5.98 cm and 4.95 to 6.35 cm, the ratio of leaf length to leaf width range is from 1.52 to 1.68 and 1.27 to 1.79, the petiole length range is from 2.87 to 4.24 cm and 3.19 to 4.54 cm, and the ratio of petiole length to total leaf length range is from 0.28 to 0.31 and 0.27 to 0.35 (Table S4). In the capsule morphology traits of *Ka* and *St*, the infructescence length range is from 8.82 to 10.9 cm and 7.80 to 10.73 cm, the length of the capsule valve range is from 6.47 to 7.18 mm and 6.49 to 8.21 mm, the width of the capsule valve range is from 3.61 to 3.89 mm and 3.00 to 3.84 mm, and the ratio of length to width of the capsule valve range is from 1.82 to 1.88 and 1.93 to 2.42. The capsule split numbers of 2-, 3-, and 4-valve are 7.82%–24.23% (*Ka*) and 0.34%–3.73% (*St*), 4.48%–77.35% (*Ka*) and 35.8%–53.11% (*St*), and 8.57%–37.69% (*Ka*) and 45.60%–63.83% (*St*). In summary, there are some differences in leaf morphological traits among different sample sites, but it lacks taxonomic value for distinguishing the two taxa. Compared with the leaf morphology traits, the capsule morphology traits have higher differentiation between sample points or between taxa. Among them, the ratio of the length to width of the capsule valve and the capsule split number is significantly different between the two taxa, which can be used as the key to distinguish them (Figures 2 and S4).

According to the systematic cluster analysis (average method) for morphological traits of 14 sample sites, *Ka* and *St* were obviously divided into two categories (Figure 2K). In addition, we made a canonical discriminant analysis using these data, and the results showed that there were significant differences in the ratio of the length to width of the capsule valve and the capsule split number. Therefore, discriminant analysis was carried out again based on the two morphological traits after screening, and the result was exactly the same as that of cluster analysis. All 14 samples are divided into two categories, and the accuracy of discrimination is 100% (Table S6).

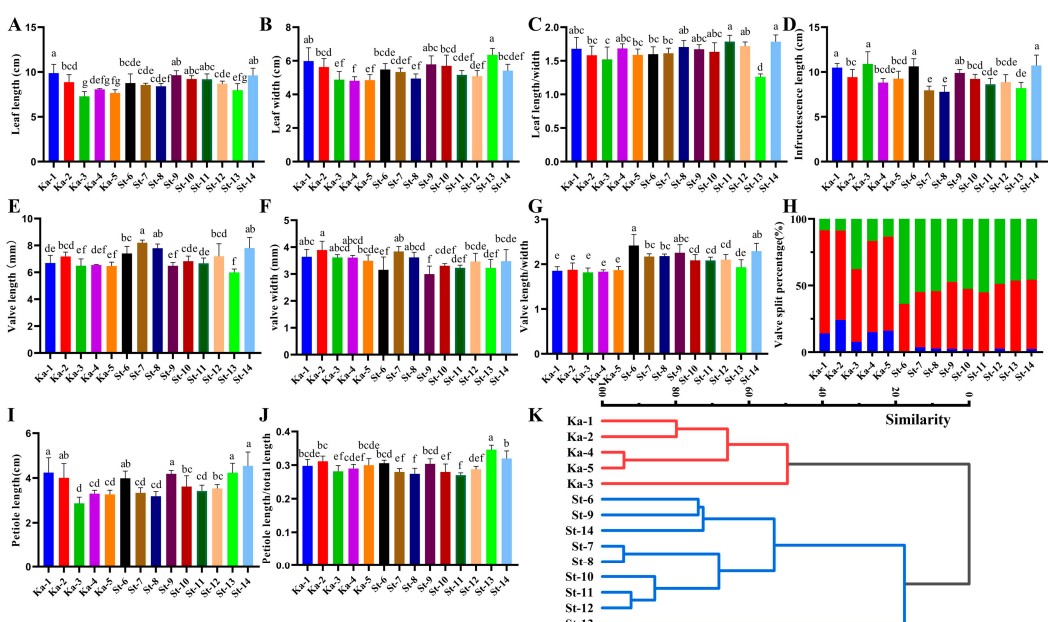

**Figure 2.** Comparison of morphological indices among different sample sites of *P. kangdingensis* (*Ka*) and *P. schneideri* var. *tibetica* (*St*). (**A**) Leaf length; (**B**) leaf width; (**C**) ratio of leaf length to leaf width; (**D**) infructescence length; (**E**) valve length; (**F**) valve width; (**G**) ratio of valve length to valve width; (**H**) valve split percentage; (**I**) petiole length; (**J**) ratio of petiole length to total leaf length; and (**K**) morphological clustering. Different lowercase letters represent significant differences between different groups, $p < 0.05$.

### 3.3. Phylogenetic Analysis

Based on the previous investigation of poplars of Sect. *Tacamahaca* and Sect. *Leucoides* in QTP, we collected the corresponding materials for phylogenetic relationship analysis (Table S2). The phylogenetic tree results showed that *St* had the closest genetic relationship with *Ka* but was far from Sc and not in the same subgroup (Figure S5A). Through the analysis of genetic structure, we found that when *K* was 8, 10, and 14, respectively, their corresponding *CV* error rates were lower (Figure S5B), indicating that the genetic structure models corresponding to the three *K* values were consistent with the actual data. When *K* = 8, St and Ka are hybrids, and their genetic components come from *Xi* (or *Ha*) and *Si*; when *K* = 10, the result of genetic structure is consistent with *K* = 8; but when *K* = 14, St becomes homozygous status, while *Ka* still shows that it is a hybrid between *Xi* (or *Ha*) and *Si* (Figure S5C). There is no relationship between *Sc* and *St*, so there will be no further analysis of *Sc*.

In order to further explore the phylogenetic relationship between *Ka* and *St* and their parent groups, we selected and analyzed *Ka*, *St*, *Si*, *Xi*, and *Ha*. The results of ML phylogenetic trees showed that *Ka* and *St* were distributed between *Si* and *Xi* (or *Ha*), and the individuals within the two taxa are grouped into two separate clusters (Figure 3A). The result of PCA is consistent with that of the ML phylogenetic tree, but only the first principal component has a higher interpretation rate (48.02%). The two hybrids were distributed among their possible parents and could not be distinguished in principal component 1, indicating that the genetic components of *Ka* and *St* were very similar (Figure 3B). Based on the cross-validation error rate, it was found that *K* = 2 might be the most suitable gene structure composition. Furthermore, the results showed that *Si* was a homozygous state, while all populations of *Ha* were heterozygous states, a few populations of *Xi* were heterozygous states, and the homozygosity was higher than that of *Ha*. Therefore, *Xi* and *Si* should be the parents of the two hybrids (*Ka* and *St*). In addition, although the genetic structure and composition of *Ka* and *St* were the same, there were some differences in the proportion of genetic components in their gene. Among them, 59% of the genetic

composition of *Ka* comes from *Xi*, 41% comes from *Si*, 63% genetic composition of *St* from *Xi*, and 37% comes from *Si* (Figure 3C).

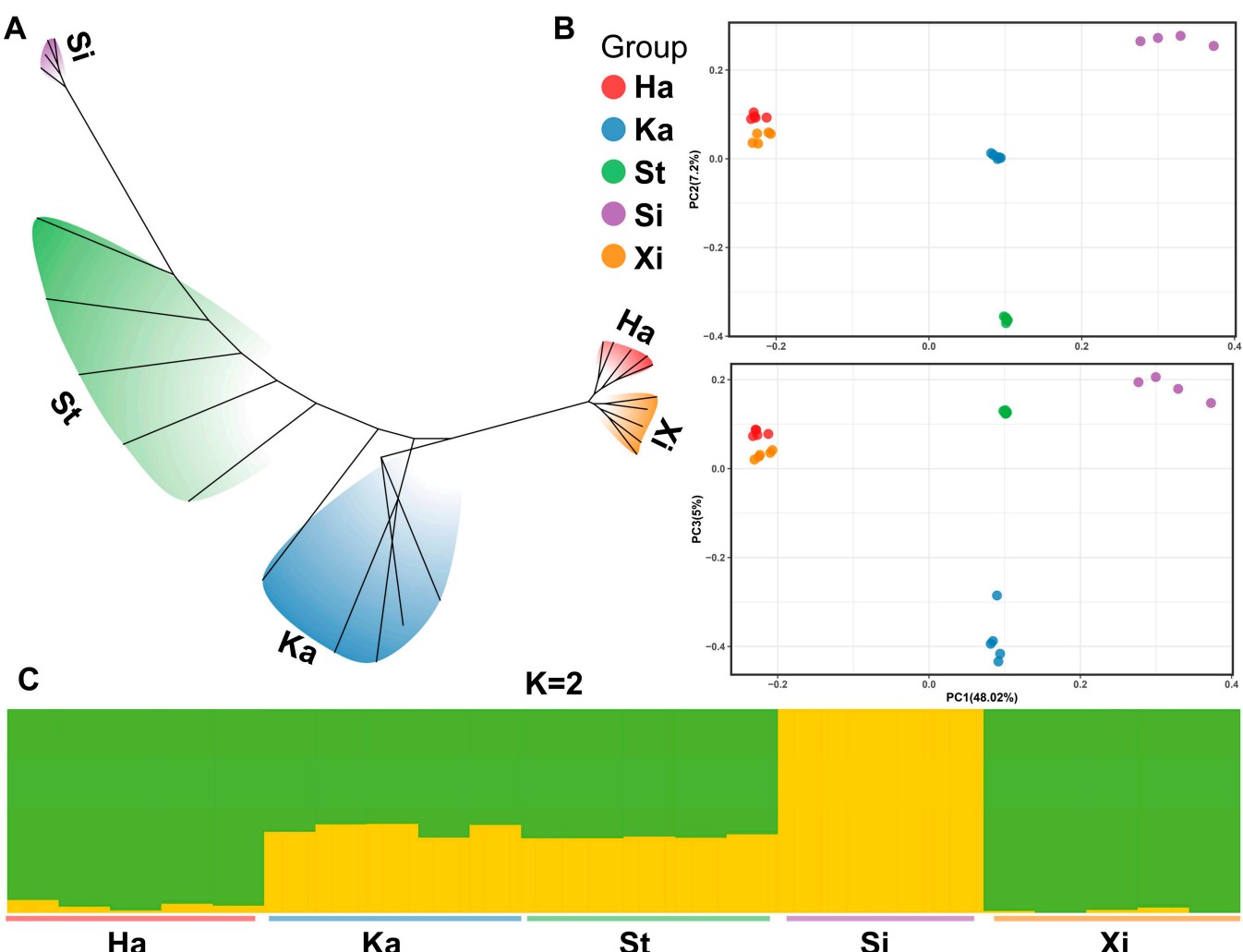

**Figure 3.** Phylogenetic relationship among *P. kangdingensis*, *P. schneideri* var. *tibetica*, and related taxa. (**A**) ML phylogenetic tree; (**B**) principal component analysis; and (**C**) genetic structure of *K* = 2. *P. haoana* (*Ha*), *P. kangdingensis* (*Ka*), *P. schneideri* var. *tibetica* (*St*), *P. simonii* (*Si*), *P. xiangchengensis* (*Xi*).

### 3.4. Gene Flow Detection and Genetic Differentiation Analysis

The results of ABBA-BABA analysis showed that all *D* statistics values were significantly greater than 0, indicating that there was significant gene flow between *Ka*, *St*, and their parents, and they may be hybrid taxa (Figure 4A). The result of the hybridization test based on HyDe software was consistent with that of the ABBA-BABA analysis. Significant hybridization signals were detected between *Ka*, *St*, and their parents, and the genetic components of the two hybrids were the same. The results showed that 41% of the genetic composition came from *Si* and 59% from *Xi* (Figure 4B).

Furthermore, the gene flow event was estimated by TreeMix software, and the residual value was between −43.1 SE and 43.1 SE when *m* = 0, indicating that there were gene flow events among these populations, mainly between *Si-Ka*, *Ka-Xi*, *St-Xi*, *Wi-Si*, and *Wi-Xi* (Figure S6). The gene flow event from *Si* to *Ka* was detected when *m* = 1. When *m* = 2, gene flow events from *Si* to *Ka* and *Wi* were detected. When *m* = 3, gene flow events from *Xi* to *Ka* and *St*, and from *Xi* to *Wi* were detected (Figure 4C), and the corresponding residual value is 0, indicating that all possible gene flow events have been detected (Figure S6). To sum up, both *Ka* and *St* are hybrid progenies of *Xi* and *Si*, and the two parents still have

continuous gene flow with *Ka* and *St*, and the gene flow from *Xi* to *Ka* was stronger than that from *Xi* to *St* (Figure 4C).

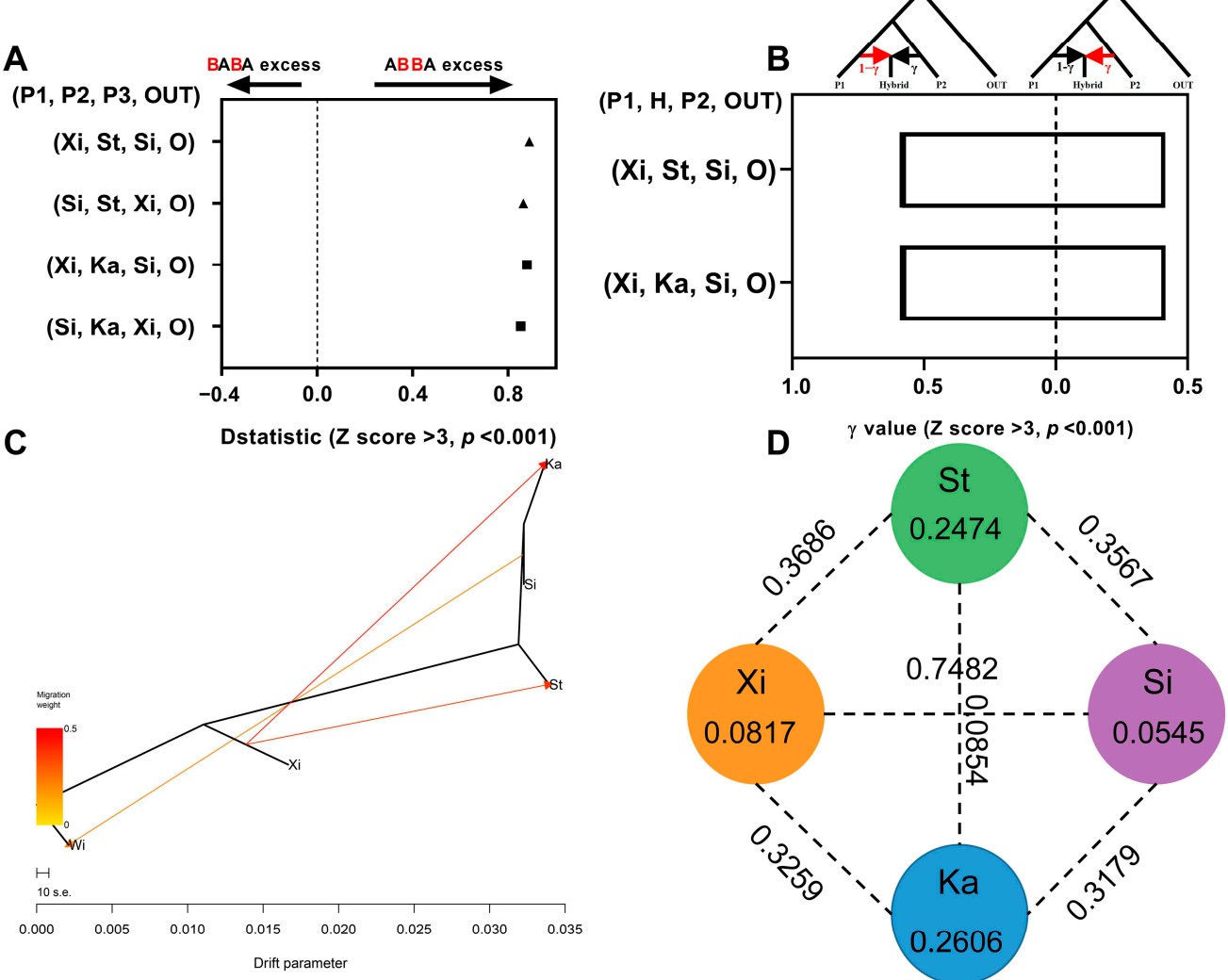

**Figure 4.** Gene flow and genetic differentiation analysis. (**A**) ABBA-BABA analysis; (**B**) hybridization detection based on HyDe; (**C**) detection of gene flow events based on TreeMix; and (**D**) nucleotide diversity ($p_i$) and relative differentiation coefficient ($F_{st}$). *P. simonii* (*Si*), *P. xiangchengensis* (*Xi*), *P. kangdingensis* (*Ka*), and *P. schneideri* var. *tibetica* (*St*).

In addition, we detected the $p_i$ of each taxon and $F_{st}$ between two taxa (Figure 4D). The results showed that the $p_i$ of *St* (0.2474) and *Ka* (0.2606) was higher, while that of *Xi* (0.0817) and *Si* (0.0545) was lower, and the difference was significant between hybrids and parents. Furthermore, the $F_{st}$ between the two species ranges from 0.0854 (*St-Ka*) to 0.7482 (*Xi-Si*), indicating that there is a large differentiation between *Xi* and *Si*, but there is almost no differentiation between *St* and *Ka*.

### 3.5. Homology Relationship Detection

We tested the absolute differentiation coefficient ($D_{xy}$) between *Ka*, *St*, and their parents. The results showed that there was little difference in $D_{xy}$ between *Ka-Si*, *Ka-Xi*, *St-Si*, *St-Xi*, and *Ka-St*, and it was evenly distributed on each chromosome (Figures 5A and S7). Because $D_{xy}$ reflects the nucleotide differences between populations, they all belong to the populations within the Sect. *Tacamahaca*, so the genetic relationship is relatively close. This

result is consistent with the result of the NJ phylogenetic tree, which makes the $D_{xy}$ of each group similar.

Furthermore, we analyzed IBD between *Ka*, *St*, and their parents, and the results showed that the shared IBD blocks between *Ka* and *Si* were much less than those between *Ka* and *Xi*. The same trend was observed in *St*, and the total number and length of shared IBD blocks of *Ka-Xi* were much larger than those of *St-Xi*. Due to the effect of gene recombination, the shared IBD blocks were broken into small fragments and could not be detected, so the shared IBD blocks of the two hybrids were less, while they shared more IBD blocks with *Xi*, and the shared IBD blocks of *Ka-Xi* was much more than that of *St-Xi* (Figure 5B,C). It may be due to the stronger gene flow of *Ka-Xi* and *St-Xi* than *Ka-Si* and *St-Si*, and the gene flow intensity between *Xi-Ka* is greater than that between *Xi-St* (Figure 4C). In addition, there are a few common IBD fragments of *Ka-St* and *Xi-Si* (Figure 5D), which may be caused by gene flow or incomplete differentiation among these species in Sect. *Tacamahaca*.

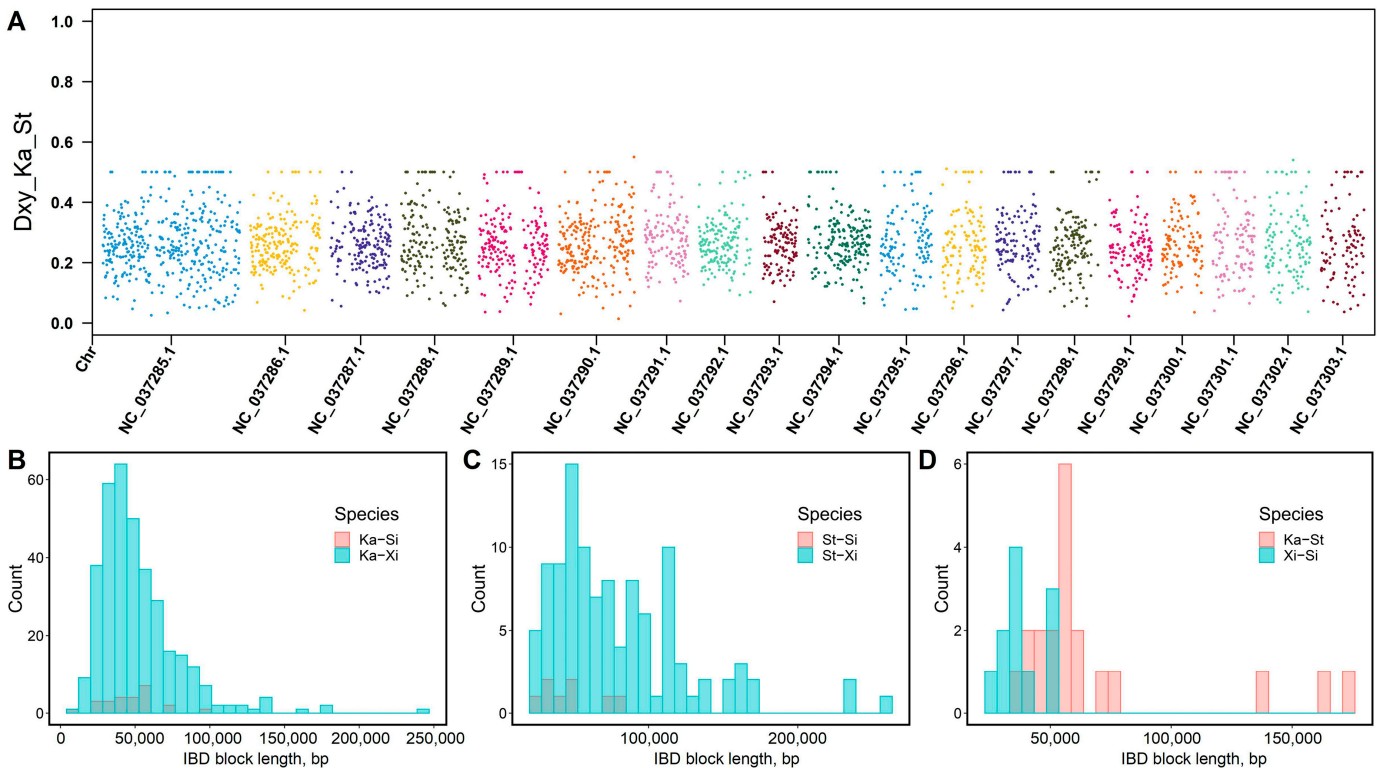

**Figure 5.** Homology analysis. (**A**) Absolute differentiation coefficient ($D_{xy}$) between *P. kangdingensis*, *P. schneideri* var. *tibetica*. (**B–D**) Shared IBD blocks between two taxa. *P. simonii* (*Si*), *P. xiangchengensis* (*Xi*), *P. kangdingensis* (*Ka*) and *P. schneideri* var. *tibetica* (*St*).

*3.6. Reconstruction of Population History*

We first simulated the effective population historical size of each individual in the four taxa by using PSMC, and the results showed that there was a great difference in the changing trend of the effective population historical size between the two hybrid progenies and their parents (Figure 6A). Among them, the changing trend of the historical effective population size of *Xi* and *Si* was about the same. At approximately 1 Mya, the population began to expand until approximately 0.4–0.5 Mya; the population size reached a peak, and then the population began to shrink rapidly. The population size of *Xi* did not slowly enter a steady state until approximately 0.02 Mya. On the other hand, the population of *Si* entered a stable period earlier than that of *Xi*, and was in a stable stage about 0.03–0.1 Mya, but then the population shrank rapidly.

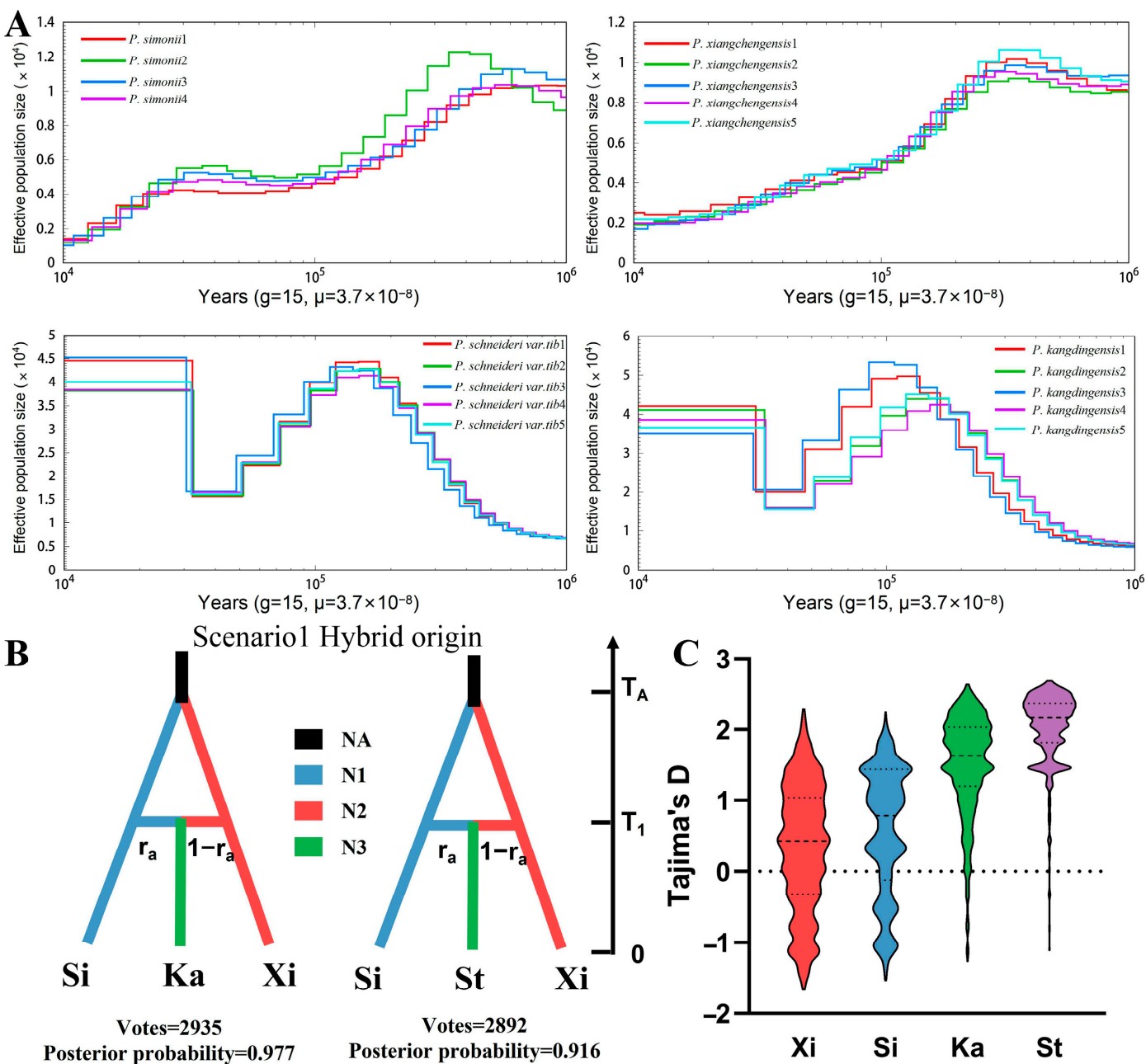

**Figure 6.** Population historical dynamics analysis. (**A**) Historical effective population size of each individual was simulated by PSMC. (**B**) The mode of origin of species is simulated by DIYABC-RF. (**C**) Tajima's *D* is based on whole genomic of four taxa. *P. simonii* (*Si*), *P. xiangchengensis* (*Xi*), *P. kangdingensis* (*Ka*), and *P. schneideri* var. *tibetica* (*St*).

The changing trend of the historical population size of *Ka* and *St* was relatively consistent. Their population began to expand rapidly approximately 1 Mya, until approximately 0.1–0.2 Mya; the population size reached a peak, and after entering a short stationary period, the population size began to shrink rapidly until about 0.05 Mya. After that, the population entered a stable period again, until about 0.03 Mya; the population size began to expand rapidly and then entered a stable period (Figure 6A). The climate fluctuation during the Quaternary Pleistocene led to the periodic cycle of population decline and post-glacial expansion, which is a characteristic of the biogeographic history of the Qinghai–Tibet Plateau flora. The difference is that the population size of *Xi* and *Si* finally entered a shrinking stage, while the population size of the hybrids rebounded rapidly during the LGM period (about

22 Kya), indicating that they had strong persistence and resilience to the Quaternary ice age and had stronger adaptability than their parents. On the high-altitude Plateau where *Ka* and *St* live, because the habitat is dry and cold, the population of parents can hardly be found. We speculate that they may not be able to adapt to the dry and cold habitat but slowly escape from the area and live in a more adaptive environment. Because of heterosis, hybrids can better adapt to the environment and survive in this habitat, but they also suffer from the great pressure of natural selection because their Tajima's *D* value is much larger than that of their parents and deviates significantly from 0 (Figure 6C).

In addition, we simulate the mode and time of species formation by DIYABC-RF software. In the three origin scenarios, scenario 1 of *Ka* (votes = 2935, posterior probability = 0.977) and *St* (votes = 2935, posterior probability = 0.977) gets the most votes (Figure S9), and the simulated data set of scenario 1 is more consistent with the actual data (Figure S8). This also proves once again that *St* and *Ka* originated from hybridization, and their parents are *Si* and *Xi*, in which *Si* and *Xi* crossed to form *Ka* about 0.108 Mya (90% CI: 0.029–0.236 Mya), and formed *Si* about 0.170 Mya (90% CI: 0.039–0.431 Mya; Figure 6B and Table S5). The time of these hybridization events is relatively earlier than that of PSMC, which may be due to the fact that the time estimated by the DIYABC-RF software is a point evaluation, and the recent gene flow signals will mask the ancient hybridization signals, thus making the estimation time earlier. The model also estimates their genetic composition. Among them, 45% (90% CI: 33%–59%) of the components of *Ka* come from *Si*, and 55% of the components of *Ka* come from *Xi*; 48% (90% CI: 43%–52%) of the components of *St* come from *Si*, and 52% of the components of *St* come from *Xi* (Table S5). Although there are some differences between the results of the genetic proportion estimated by point evaluation of DIYABC-RF and HyDe (or Structure analysis), which may be caused by different algorithms of different software, their overall trend is consistent, showing that more *Xi* genes are inherited in *Ka* and *St*, which is consistent with the results of gene flow and IBD (Figures 4C and 5B).

*3.7. Positive Selection Analysis*

In order to verify that the hybrid population has stronger environmental adaptability than their parent population, we screened the selected intervals in the hybrids based on the large $F_{st}$ between the hybrid offspring and their parents. We carried out functional annotation and enrichment analysis of the genes in these intervals based on GO and KEGG databases. First of all, according to the first 5% $F_{st}$ values, 35 and 76 candidate genes were screened between *Ka*, *St*, and two parents (Figure 7A–D), respectively, and then the screened genes were enriched by GO and KEGG databases. The result of KEGG analysis showed that it was mainly enriched in the processes of ABC transporters, Valine, leucine and isoleucine degradation, Pantothenate and CoA biosynthesis, SNARE interactions in vesicular transport, Propanoate metabolism, Nitrogen metabolism, and biosynthesis of secondary metabolites, which was closely related to stress resistance and growth (Figure 7C). Moreover, the biological process of GO annotation showed that most of the genes were a response to stress (Figure 7D). The specific functions of stress-related genes were further analyzed, and the results showed that these genes were mainly involved in abiotic stress (salt stress, osmotic stress, hypoxia stress, drought stress, and cold stress) and biotic stress (fungi stress and bacteria stress). In addition, these genes were also involved in the regulation of plant growth and development (Table S7).

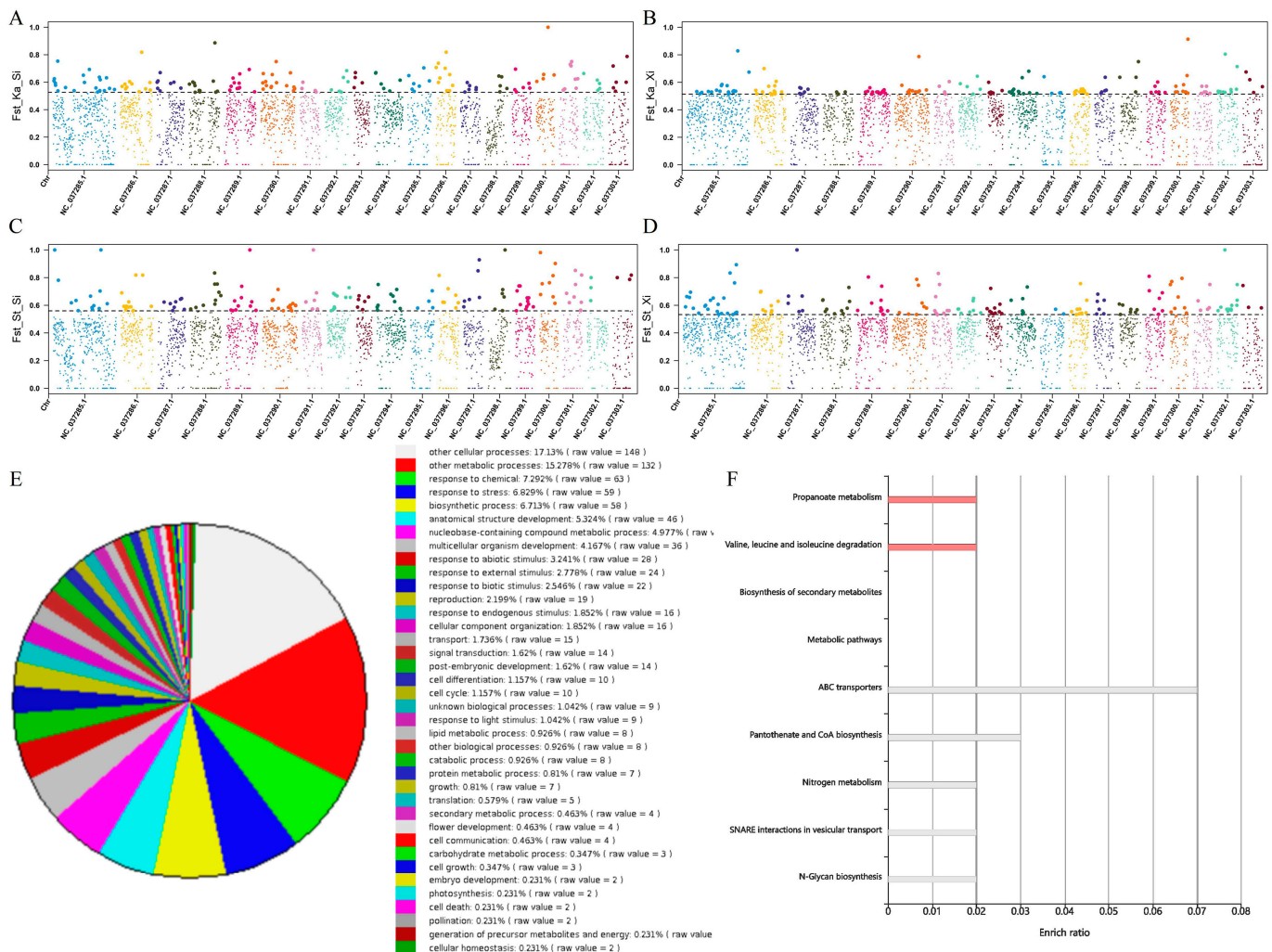

**Figure 7.** Gene screening and annotation. (**A–D**) $F_{st}$ values between *Ka*, *St* and their two parents. The black line is the threshold line of the top 5%. (**E,F**) GO and KEGG annotation of candidate genes.

### 3.8. Prediction of Suitable Distribution Area

In order to further evaluate the adaptability of *Ka* and *St* populations, we used the climatic data from four periods to predict their suitable distribution areas. The AUC values of the prediction models in the four periods are all greater than 0.9 (Figure S10), indicating that the model can fit the data accurately. In the LIG period, the population was mainly distributed in the west and southeast of Sichuan, and the optimal distribution area was small (Figure 8A). In the LGM period, the population began to expand to the southwest (Figure 8B), and the optimal distribution area increased obviously, which was completely consistent with the result predicted by PSMC. Currently, the optimal distribution area has shrunk sharply (Figure 8C), which is basically consistent with the scope of our investigation. Moreover, we also predicted the change of the population in the future, and the results showed that the optimal distribution area will still be in western Sichuan, centering on it and expanding in all directions (Figure 8D). This is very consistent with the results predicted by the whole genome data, which shows that the hybrid has better adaptability, and its suitable distribution area expands gradually in the future.

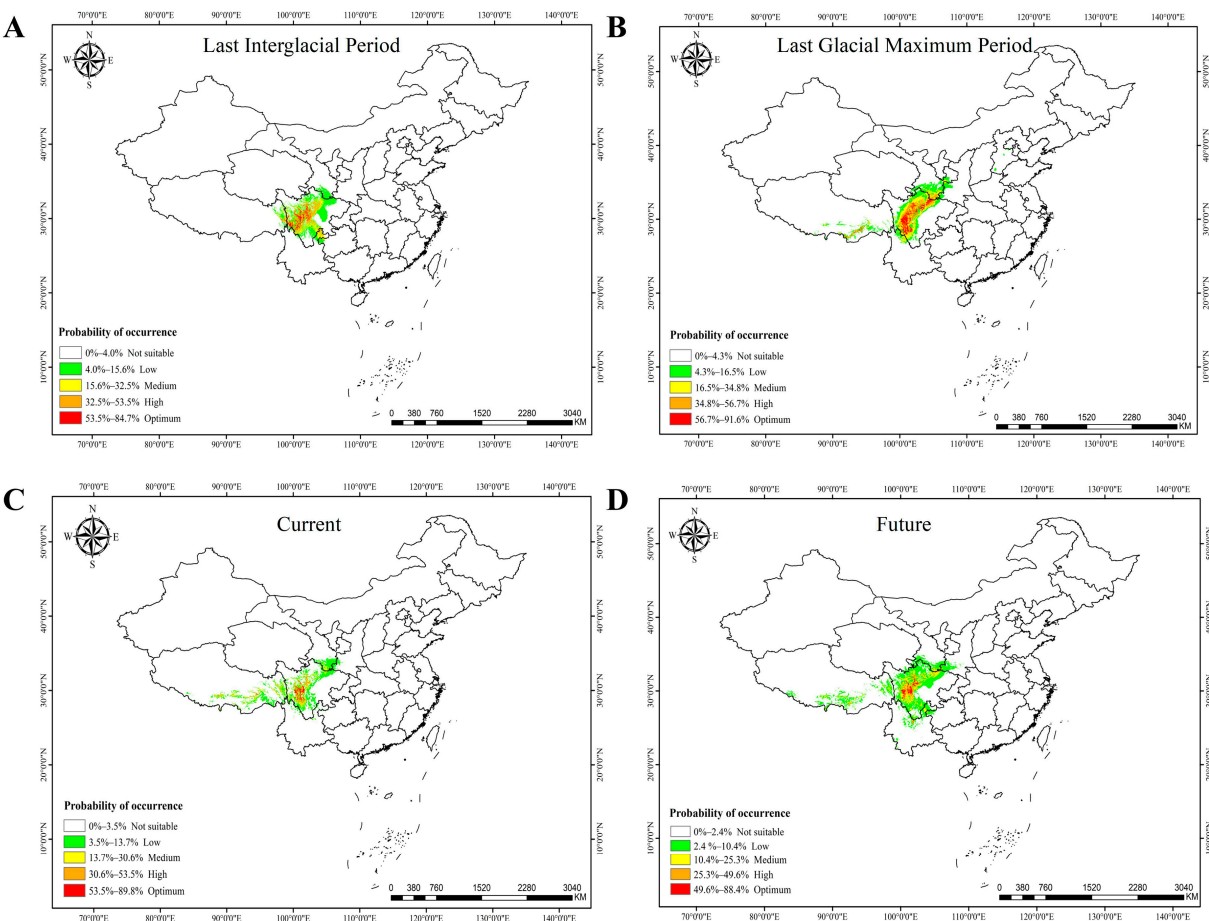

**Figure 8.** Prediction of suitable distribution areas of *Ka* and *St* in different periods. (**A**) The Last Interglacial period (LIG); (**B**) the Last Glacial Maximum period; (**C**) current; and (**D**) 2061–2080 years (Future).

## 4. Discussion

The identification of species boundaries is very important for the development and utilization of species and biodiversity conservation [42]. However, there are extensive crosses among species of *Populus* [37,43,44], resulting in a large number of morphological variations, which makes it extremely difficult to identify species of the genus by morphological traits. With the rapid development of next-generation sequencing technology, integrated phylogeny has been widely used in the study of species [45–47]. It has relatively high objectivity and maneuverability to define species by comprehensive means.

For the definition of *Ka* and *St* populations, we provide three strong pieces of evidence to support that *Ka* and *St* are the same taxa, so we should cancel the taxonomic status of *St* and merge it into *Ka*. First of all, *Ka* and *St* are distributed on the Plateau, the habitat is very similar, and the distribution range is narrow; only Sichuan Province has a stable population (Figure 1). Secondly, based on the comparison of the morphological traits of leaves and capsules, the morphological traits of leaves had little difference between the two taxa, so it was impossible to distinguish them (Figures 2, S2 and S4). The same results were obtained in the common garden experiments (Figure S3). In the morphological traits of the capsules, only two traits (the ratio of length to width of capsule valve and the capsule split number) had significant differences between the two taxa (Figures 2 and S4). In addition, although the two taxa can be distinguished by discriminant analysis and systematic cluster analysis, the difference between them is only shown in the above two traits. Finally, based on the analysis of genome-wide resequencing data, the phylogenetic relationship between *Ka* and *St* is very close, and the source of genetic components is exactly the same, with only a small

difference in the proportion of genetic components (Figure 3). Moreover, the $F_{st}$ between them is extremely small (Figure 4D), indicating that they are the same taxa. Therefore, morphological variation within a certain range can not be used as an index to define a species but should be explored comprehensively through the geographical distribution, morphological traits and molecular genetic data of a species in order to obtain objective and accurate results.

Hybridization is an important process in biological evolution, which is ubiquitous in nature and creates a complex evolutionary network for the tree of life [48–50]. Hybridization exists in 16%–34% of families and 6%–16% genera of plants in nature [51], which can give plants richer variation, stronger adaptability and competitiveness, and make them dominant in community regeneration [52,53]. Firstly, based on the morphological traits and geographical distribution, we inferred that *Ka* and *St* are hybrids, and *Si* is one of its parents. Both of them have the typical traits of *Si*, such as the capsule being 2-valve, long ovate, the sprouting branches and leaves with weak growth are inverted wedges, and the seedlings of them are very similar to *Si* (Figures S2 and S3). With the advent of the genome era, many outstanding problems have been broken through, which can further reveal the process and mechanism of hybrid speciation while identifying natural hybridization. Therefore, we also carried out a variety of analyses (ABBA-BABA, HyDe, TreeMix, ABC model and IBD) based on the whole genome SNPs to prove that *Ka* and *St* originated from the hybridization of *Xi* and *Si*, and there was a strong gene flow from Xi to them (Figures 4, 5 and 6B).

Furthermore, through PSMC analysis, it was found that the effective historical population size of parents and hybrids all had a contraction period (Figure 6A), which was related to the drastic change of Quaternary glacial climate on the QTP, and this phenomenon existed in many organisms, such as walnut [54], ironwood tree [55] and birch [56] and panda [57]. However, because the populations of *Ka* and *St* were endowed heterosis, their population size rebounded rapidly during the glacial period (Figure 6A), which indicated that they had strong persistence and resilience compared with their parents during the Quaternary glacial period. Moreover, we also found a large number of genes related to stress resistance and growth regulation in hybrids based on positive selection analysis (Figure 7). It is possible that these genes make *Ka* and *St* stronger adaptability, and the suitable distribution area of the taxa may gradually expand in the future (Figure 8D).

## 5. Conclusions

In this study, based on the extensive and long-term investigation of the *Populus* on the QTP and its surrounding areas, we found two taxa of Plateau poplars (*Ka* and *St*) with great cultivation value. They are very similar in morphological traits and habitat types, and the distribution range is very narrow, only found on the Plateau surface from 3400 to 4000 m above sea level in western Sichuan. First of all, through the description and statistics of morphological traits, there are only a few differences between the two taxa. Secondly, the phylogenetic analysis based on the whole genome resequencing data revealed that *St* was not related to *Sc*, but very homologous to *Ka*; they all originated from the hybridization of *Xi* and *Si*, and *Xi* still maintained a stronger gene flow with them than *Si*. In addition, the relative differentiation coefficient between them is very small, so *St* should be classified into *Ka* as the same taxa. Finally, based on the comprehensive analysis, it is revealed that the fluctuation of their population size is closely related to the Quaternary glacial-interglacial cycle, showing stronger adaptability than their parents, and the population suitable distribution areas will gradually expand in the future.

**Supplementary Materials:** The following supporting information can be downloaded at: https://www.mdpi.com/article/10.3390/f14051003/s1, Figure S1: Field investigation area; Figure S2: Habitats and specimens of *P. schneideri* var. *tibetica* and *P. kangdingensis*; Figure S3: Cutting seedlings of *P. schneideri* var. *tibetica* and *P. kangdingensis*; Figure S4: Comparison of morphological traits between two taxa; Figure S5: Phylogenetic relationship of poplar on the Qinghai-Tibet Plateau; Figure S6: The ML phylogenetic tree with migration events; Figure S7: The distribution of absolute differentiation coefficients ($D_{xy}$) between two populations on the whole genome; Figure S8: Simulated

data projection and model error rate of three scenarios; Figure S9: The three possible origin models of two taxa; Figure S10: Evaluation of suitable distribution areas by MaxEnt software of poplar (*Ka* and *St*) in four periods; Table S1: The basic information of sample sites of *P. kangdingensis* and *P. schneideri* var. *tibetica*; Table S2: *Populus* material for whole genome resequencing; Table S3: A statistical summary of whole genome resequencing data from 106 individuals of 22 taxa; Table S4: Comparison of morphological traits among different sample sites of *P. kangdingensis* and *P. schneideri* var. *tibetica*; Table S5: Posterior mean, median (i.e., 50% quantile) as well as 5% and 95% quantiles (and hence 90% credibility interval) of each parameter in the hybrid origin model for *P. kangdingensis* and *P. schneideri* var. *tibetica*; Table S6: Classification results of 14 sample points based on canonical discriminant analysis; Table S7: Functional annotations of the differentiated genes between hybrid taxa and their parents.

**Author Contributions:** X.-Q.W. and Y.-J.S. conceived and performed the original research project. Y.-J.S., J.-X.M., J.-L.H., F.H., L.-H.C. and X.-Q.W. participated in the field investigation, material collection and common garden management. X.-Q.W. and Y.-J.S. carried out data and bioinformatics analyses. Y.-J.S. and X.-Q.W. wrote the paper. X.-Q.W. supervised the experiments. F.H. and X.-Q.W. revised the writing. X.-Q.W. and L.-H.C. obtained the funding for the research project. All authors have read and agreed to the published version of the manuscript.

**Funding:** This work was supported by the National Natural Science Foundation of China (31870645 to X.-Q.W.) and supported by Sichuan Science and Technology Program (2021YFYZ0032 to L.-H.C.).

**Data Availability Statement:** All data necessary for interpretation and building upon the methods can be found in the article tables and electronic Supplementary information.

**Acknowledgments:** We thank all the members (Y.H. Ding, C.L. Wang, C. Jia, S.L. Zhang, Z. Li, Z.S. Sun, H. Yang, J.Q. Song, B. Luo, S.J. Jiang, W. Wang, and F.Y. Wu) who contributed to the field investigation in the early stage of this study.

**Conflicts of Interest:** The authors declare no conflict of interest.

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
