# Peer review of "Study on the Origin and Classification of Two Poplar Species on the Qinghai–Tibet Plateau"

_forests, doi:10.3390/f14051003_

Round 1
Reviewer 1 Report
The manuscript presents interesting evaluation of the taxonomy status of two poplar species from Sichuan – P. kangdingensis and P. schneideri var. tibetica using a broad array of morphological traits and genome resequencing. Many species, not only within the genus Populus but also of other genera, were described based on morphological traits of assimilatory (leaves) or generative organs (flowers, fruits, infrutescenses, seed) require the verification of their taxonomic status by sophisticated methods of molecular taxonomy. It is also the case of above mentioned two poplar species from Sichuan.
I found the manuscript too long, not easy to follow the text, both in the introductory part and also describing the results of morphometry and molecular analyses. I recommend to shorten the Introduction and to state clearly the aim of the study. The statement “Therefore, based on the comprehensive study of morphological taxonomy, ecological geography and molecular systematics, we focus on the classification and origin of P. kangdingensis and P. schneideri var. tibetica, and further verify the rationality of their classification and treatment.” is rather vague and the working hypotheses are hidden the sentence.
Material and methods. I would expect to describe clearly the size of natural range of both species. Of course, it is possible to recalculate the approximate size of sampling are from the coordinates of sampling sites, but it is necessary to show the size of entire natural range and the proportion of sampling sites area. I am also missing the description of the data matrix in detail (14 populations × 5 trees × 50 leaves ?), how many trees were used for morphometry of leaves and capsules and/or how many trees were used for resequencing. Statement “After testing the concentration and purity of the extracted DNA samples, the DNA samples with high purity and integrity were selected and sent to Novogene Co., Ltd. (Beijing) for genome resequencing, variant calling and annotating, then the high-quality SNPs was used for follow-up analysis.” is telling nothing about the size of data array.
Statistical analysis of data of morphological traits describe the use of Student´s t-test and ANOVA. I wonder why the authors did not use some method of multivariate statistical analyses e.g. canonical discriminant analysis. Except that, I have some doubts about the general sample size. Was it enough to use only five trees per site and was it necessary to evaluate 50 leaves per tree? I am also missing the description of cluster analysis (similarity or distance matrix and model of cluster analysis).
Results. When describing the results of morphometry, the authors used rather unconventional expressions. Instead of “the leaf length is between 7.28~9.88 cm, the leaf width is between 4.81~5.98 cm” I would recommend to use the expression “the leaf length range is between 7.28 and 9.88 cm” or preferably “the leaf length range is from 7.28 to 9.88 cm”. Except that, using the symbol “~” is not correct at these sentences.
The author should take care for proper using of “hyphen”. For symbols as minus, from–to (e.g. page range) and inserted sentence N-dash should be used. There are many places e.g. lines 378–381 where this inconsistency is confusing.
Discussion. I recommend to remove the first half of the first paragraph in Discussion. It is not linked with the results and except that there were other hundreds of genera (species) where morphometry was used. Shortening of the text is possible also in the next paragraphs. The writing of the Discussion would be easier if the working hypotheses and aims would be defined more precisely.
References. Unfortunately, the authors did not follow the Instructions for Authors, both what concerns the citation in text and also the list of references. All cited references in text are in the list of references. The references are ordered as they appeared in the text. It is necessary to replace the cited references by consecutive numbers and correct all formatting imperfections in the list of references. In the list of references there is one item doubled (Echenwalder, J. and Eckenwalder, J.) and after removal one of them the numbering in the list of references should be altered.
Summarizing the pros and cons of the submitted manuscript I would like to recommend:
· to shorten the Introduction especially the first two paragraphs)
· to clearly formulate the aims and working hypothesis
· to use appropriate methods to answer the aims and working hypotheses
· to discuss the results according to the aims and working hypothesis (remove redundant paragraphs and quotations).
The manuscript needs major revision.

Author Response
Dear Editor:
We would like to thank you and the Reviewers for your valuable comments and suggestions that helped us further our understanding in this field as well as improved our manuscript. After carefully considering the reviewers’ comments, we have revised our manuscript point-by-point. For the major comments, we had responded to each as listed below. For your convenience, all changes made in the manuscript were tracked.
Please do not hesitate to contact me in case the manuscript has any further issues.
Thank you very much for your help.
We are looking forward to hearing from you soon.
Yours sincerely,
Xue-Qin Wan, Ph.D.
Professor of forest tree genetics and breeding
College of Forestry,
Sichuan Agricultural University, Chengdou 611130, China
E-mail: [email protected]
-------------------------------------------------------------------------------------------------------
Responses to reviewer 1:
[Comment 1] I found the manuscript too long, not easy to follow the text, both in the introductory part and also describing the results of morphometry and molecular analyses. I recommend to shorten the Introduction and to state clearly the aim of the study. The statement “Therefore, based on the comprehensive study of morphological taxonomy, ecological geography and molecular systematics, we focus on the classification and origin of P. kangdingensis and P. schneideri var. tibetica, and further verify the rationality of their classification and treatment.” is rather vague and the working hypotheses are hidden the sentence.
[Our response1]: Thank you for your suggestions. We have shortened the introduction and revised the original statement to make the purpose of this study more clear.
The original statement is “Therefore, based on the comprehensive study of morphological taxonomy, ecological geography and molecular systematics, we focus on the classification and origin of P. kangdingensis and P. schneideri var. tibetica, and further verify the rationality of their classification and treatment.”, and we have revised it to “Therefore, under the background of extensive investigation on the genus Populus (22 taxa) on the QTP, the habitat and morphological data of 70 individuals from 14 sample sites of P. kangdingensis and P. schneideri var. tibetica were collected, and 10 individuals of them were randomly selected for whole genome resequencing to further reveal their classification and origin. ” (in line 86–90)
[Comment 2] Material and methods. I would expect to describe clearly the size of natural range of both species. Of course, it is possible to recalculate the approximate size of sampling are from the coordinates of sampling sites, but it is necessary to show the size of entire natural range and the proportion of sampling sites area.
[Our response 2]: Thank you for your suggestions. We have supplemented the natural distribution of the population in Figure 1. Furthermore, in order to show the approximate size of entire natural distribution of the proportion, we simulated the natural distribution suitable area of the species in four periods (LIG, LGM, Current and Future) by using MaxEnt software. This contents are supplemented in the “Materials and Methods” and “Results” in the paper.
“Prediction of suitable distribution area of species
Firstly, 19 global climate factors of four periods (the Last Interglacial, the Last Glacial Maximum, Current and 2061–2080 year) are downloaded from the WorldClim database (https://www.worldclim.org/), and their spatial resolutions are all 2.5 m. In order to avoid the over-fitting of the model caused by multiple collinearity among environmental factors, according to the contribution rate of model training and the results of Pearson correlation analysis from environmental factors in SPSS, we eliminate the environmental factors whose correlation coefficient is higher than 0.8 and the contribution rate is small. Then, combined with the coordinate information of the sample points of Poplar, the fitting analysis of the suitable distribution area was carried out by using MaxEnt v.3.4.3 [42]. The 25% distribution data is selected as the detection data of the model, 75% data is used as training data, repeated 10 times, and the other parameters are set as default values for modeling. Finally, based on the fitted mean value, ArcGIS v.10.8 is used to visualize the suitable distribution area.” (in line 284–297)
“Prediction of suitable distribution area
In order to further evaluate the adaptability of Ka and St populations, we used the climatic data of four periods to predict their suitable distribution areas. The AUC values of the prediction models in the four periods are all greater than 0.9 (Fig. S10), indicating that the model can fit the data accurately. In the LIG period, the population was mainly distributed in the west and southeast of Sichuan, and the optimal distribution area was small (Fig. 8A). In the LGM period, the population began to expand to the southwest (Fig. 8B), and the optimal distribution area increased obviously, which was completely consistent with the result predicted by PSMC. At current, the optimal distribution area has shrunk sharply (Fig. 8C), which is basically consistent with the scope of our investigation. Moreover, we also predicted the change of the population in the future, and the results showed that the optimal distribution area will be still in western Sichuan, centering on it and expanding in all directions (Fig. 8D). This is very consistent with the results predicted by the whole genome data, which shows that the hybrid has better adaptability and its suitable distribution area expands gradually in the future.” (in line 522–536)
[Comment 3] I am also missing the description of the data matrix in detail (14 populations × 5 trees × 50 leaves ?), how many trees were used for morphometry of leaves and capsules and/or how many trees were used for resequencing. Statement “After testing the concentration and purity of the extracted DNA samples, the DNA samples with high purity and integrity were selected and sent to Novogene Co., Ltd. (Beijing) for genome resequencing, variant calling and annotating, then the high-quality SNPs was used for follow-up analysis.” is telling nothing about the size of data array.
[Our response 3]: Thank you for your suggestions. We have added the details of samples acquisition and measurement in “Material and Methods”, as well as the material acquisition, sequencing process and data filtering of re-sequencing. The additions are as follows:
“In each sample site, at least 5 representative adult healthy female plants were selected as samples, and the infructescences, short branches with leaves were collected to determine the classification characteristics. More than 20 short branches with leaves under the canopy of the sample trees were selected to ensure that more than 50 typical intact leaves could be obtained of each tree (14 populations × 5 trees × 50 leaves). Due to the differences in altitude, latitude and habitat conditions of the sample sites, the capsules of trees at most sample sites cracked and the infructescences had fallen to the ground at the time of the investigation, but the capsules of some sample sites had not cracked and the infructescences had not fallen off from the tree. For the plants whose infructescences were still hanging on the tree, 50 infructescences are randomly collected, while for the plants whose infructescences had fallen off, at least 50 intact and clean infructescences were picked up under the crown (14 populations × 5 trees × 50 infructescences). The all samples of leaves and infructescences were collected and numbered, then packed in the sample bag and brought back to the room to determine the relevant traits. The corresponding cuttings of all samples were collected and brought back to the Modern Agricultural Research and Development Base of Sichuan Agricultural University (103°38′43″ E, 30°33′24″ N) for subsequent common garden experiments.” (in line 114–131)
“For leaf samples of each tree, 15 leaves were randomly selected from 50 intact leaves to measure petiole length, leaf length (excluding petiole), leaf width and total leaf length (including petiole) with a ruler. Then, according to the above traits, the ratio of leaf length to leaf width and the ratio of petiole length to total leaf length were calculated. For capsule samples of each tree, 15 intact infructescences were randomly selected from 50 infructescences to measure the length of the infructescences with a ruler, then took down all the capsules from the above 15 infructescences and placed them in an oven at 40 ℃ for 24 hours.” (in line 142–149)
“Because there are more polyphenol polysaccharides in the old leaves of poplar, the 106 individuals of 22 taxa (Table S2, including 5 individuals of Ka and 5 individuals of St in this study) were collected from the common garden, and the new young leaves of cuttings were collected for the extraction of DNA. After the young leaves were collected, they were quickly frozen in liquid nitrogen, and the whole genomic DNA of each plant was extracted by slightly modified CTAB method (1 ml mixture of Vphenol : Vchloroform : Visoamylol = 25 : 24 : 1) [17]. The 106 sample’s DNA were sent to Novogene Co., Ltd. (Beijing, China) for genome re-sequencing. After passing the quality test of DNA, the fragments with a length of 350 bp were randomly broken by a Covaris crusher and created libraries using NEB Next® Ultra™ II DNA Library Prep Kit. Then, the constructed libraries were sequenced by the Illumina NovaseqTM platform with an average sequencing depth of 10× (Table S3).
The low-quality reads of raw reads were inspected and filtered using FastQC9 and Trimmomatic v.0.40 [18]. All clean reads of each sample were mapped to the P. trichocarpa reference genome [6] using the BWA-MEM algorithm of BWA v.0.7.17 with default parameters [19]. The Sequence Alignment Map (SAM) format files were converted to Binary Alignment Map (BAM) format files using SAMtools v.0.1.19 [20]. Moreover, polymerase chain reaction duplicates were removed using Picard tools v.2.1.1. The Genome Analysis Toolkit (GATK) v.4.2.6.1 was used to call single nucleotide polymorphisms (SNPs) and short InDels with default settings from each species separately [21], and to joint SNPs from all individuals. SNP sites were filtered according to mass value, site depth, Fisher test value, and comparison quality using GATK with the parameters as “QUAL < 30 || QD < 3.0 || FS > 60.0 || MQ < 40.0,” and missing data were filtered using SAMtools with the parameters as “Dp3-miss0.2-maf 0.05” [20]. The final SNPs were annotated referring to the P. trichocarpa genome using ANNOVAR [22].
The individuals mapping to P. trichocarpa were prepared for two SNPs data sets: 1) A 22-taxa SNPs data set, which was used for phylogenetic TREE, and STRUCTURE analysis; 2) A 5-taxa SNPs data set including Ka, St, P. simonii (Si), P. xiangchengensis (Xi), and P. haoana (Ha), which was used for phylogenetic TREE, PCA, STRUCTURE, and subsequent hybridization, gene flow and population history dynamic analysis.” (in line 156–186)
[Comment 4] Statistical analysis of data of morphological traits describe the use of Student´s t-test and ANOVA. I wonder why the authors did not use some method of multivariate statistical analyses e.g. canonical discriminant analysis.
[Our response 4]: Thank you for your questions. First of all, we agree with the views of the reviewer, so we add canonical discriminant analysis in the Statistical analysis of data of morphological traits, and the results are completely consistent with the results of cluster analysis, these indicators can divide the two taxa into two categories, and the correct rate is 100%. This contents are supplemented in the “Materials and Methods” and “Results” in the paper.
“Canonical discriminant analysis was also performed through the SPSS software. In addition, the systematic clustering analysis based on morphological indicators was performed by Origin software, in which the clustering method is set as the Average method, the distance type is set as Pearson correlation, and the other parameters are default.” (in line 303–307)
“According to the systematic cluster analysis (average method) for morphological traits of 14 sample sites, Ka and St were obviously divided into two categories (Fig. 2K). In addition, we made a canonical discriminant analysis using these data, and the results showed that there were significant differences in the ratio of the length to width of the capsule valve and the capsule split number. Therefore, discriminant analysis was carried out again based on the two morphological traits after screening, and the result is exactly the same as that of cluster analysis. All the 14 samples are divided into two categories, and the accuracy of discrimination is 100% (Table S6).” (in line 339–346)
[Comment 5] Except that, I have some doubts about the general sample size. Was it enough to use only five trees per site and was it necessary to evaluate 50 leaves per tree? I am also missing the description of cluster analysis (similarity or distance matrix and model of cluster analysis).
[Our response 5]: Thank you for your questions. First of all, the distribution range of P. kangdingensis and P. schneideri var. tibetica is very narrow, only distributed in Sichuan province. In addition, when we collect samples, we mainly refer to Flora, only collect samples that are the same or similar to the original samples, while some samples with large differences are not collected. In order to avoid collecting asexual individuals, we will collect individual samples at an interval of 100 m. Therefore, the final number of individuals collected in each sample site is 5, but it can well represent the morphological characteristics of the population in this sample site. In addition, we also supplement the specific parameters of the cluster analysis method in the Methods and Materials.
As for the question of "whether it is necessary to evaluate 50 leaves per tree?", it may be our negligence that the description of the method and material is not detailed enough, leading to the misunderstanding here. In our field investigation, due to the long time of going out each time (usually about 15 days), and the determination of leaf traits needs to be brought back to the laboratory for measurement, the transition time is long, which may cause damage to some leaves. Therefore, in the field investigation, we collect as many leaves as possible from each tree in good shape (usually at least 50 leaves), and 15 leaves are randomly selected from the intact leaves to measure the morphological characteristics of each tree. In order to avoid another misunderstanding, we have made a detailed description in the methods and materials.
“In each sample site, at least 5 representative adult healthy female plants were selected as samples, and the infructescences, short branches with leaves were collected to determine the classification characteristics. More than 20 short branches with leaves under the canopy of the sample trees were selected to ensure that more than 50 typical intact leaves could be obtained of each tree (14 populations × 5 trees × 50 leaves). Due to the differences in altitude, latitude and habitat conditions of the sample sites, the capsules of trees at most sample sites cracked and the infructescences had fallen to the ground at the time of the investigation, but the capsules of some sample sites had not cracked and the infructescences had not fallen off from the tree. For the plants whose infructescences were still hanging on the tree, 50 infructescences are randomly collected, while for the plants whose infructescences had fallen off, at least 50 intact and clean infructescences were picked up under the crown (14 populations × 5 trees × 50 infructescences). The all samples of leaves and infructescences were collected and numbered, then packed in the sample bag and brought back to the room to determine the relevant traits. The corresponding cuttings of all samples were collected and brought back to the Modern Agricultural Research and Development Base of Sichuan Agricultural University (103°38′43″ E, 30°33′24″ N) for subsequent common garden experiments.” (in line 114–131)
“For leaf samples of each tree, 15 leaves were randomly selected from 50 intact leaves to measure petiole length, leaf length (excluding petiole), leaf width and total leaf length (including petiole) with a ruler. Then, according to the above traits, the ratio of leaf length to leaf width and the ratio of petiole length to total leaf length were calculated. For capsule samples of each tree, 15 intact infructescences were randomly selected from 50 infructescences to measure the length of the infructescences with a ruler, then took down all the capsules from the above 15 infructescences and placed them in an oven at 40 ℃ for 24 hours.” (in line 142–149)
“In addition, the systematic clustering analysis based on morphological indicators was performed by Origin software, in which the clustering method is set as the Average method, the distance type is set as Pearson correlation, and the other parameters are default.” (in line 304–307)
[Comment 6] Results. When describing the results of morphometry, the authors used rather unconventional expressions. Instead of “the leaf length is between 7.28~9.88 cm, the leaf width is between 4.81~5.98 cm” I would recommend to use the expression “the leaf length range is between 7.28 and 9.88 cm” or preferably “the leaf length range is from 7.28 to 9.88 cm”. Except that, using the symbol “~” is not correct at these sentences.
The author should take care for proper using of “hyphen”. For symbols as minus, from–to (e.g. page range) and inserted sentence N-dash should be used. There are many places e.g. lines 378–381 where this inconsistency is confusing.
[Our response 6]: Thank you for your suggestions. We have modified the unconventional expressions, and the symbol “~” is instead of the expression “... range is from xx to xx.” In addition, N-dash and minus have been modified in some places so that they can be clearly distinguished.
“In the leaf morphological traits of Ka and St, the leaf length range is from 7.28 to 9.88 cm and 7.97 to 9.64 cm, the leaf width range is from 4.81 to 5.98 cm and 4.95 to 6.35 cm, the ratio of leaf length to leaf width range is from 1.52 to 1.68 and 1.27 to 1.79, the petiole length range is from 2.87 to 4.24 cm and 3.19 to 4.54 cm, and the ratio of petiole length to total leaf length range is from 0.28 to 0.31 and 0.27 to 0.35 (Table S4). In the capsule morphology traits of Ka and St, the infructescence length range is from 8.82 to 10.9 cm and 7.80 to 10.73 cm, the length of capsule valve range is from 6.47 to 7.18 mm and 6.49 to 8.21 mm, the width of capsule valve range is from 3.61 to 3.89 mm and 3.00 to 3.84 mm, the ratio of length to width of capsule valve range is from 1.82 to 1.88 and 1.93 to 2.42. The capsule split number of 2-, 3-, 4-valve are 7.82–24.23% (Ka) and 0.34–3.73% (St), 4.48–77.35% (Ka) and 35.8–53.11% (St), 8.57–37.69% (Ka) and 45.60–63.83% (St). In summary, there are some differences in leaf morphological traits among different sample sites, but it lacks taxonomic value for distinguishing the two taxa. Compared with the leaf morphology traits, the capsule morphology traits have higher differentiation no matter between sample points or between taxa. Among them, the ratio of the length to width of the capsule valve and the capsule split number are significantly different between two taxa, which can be used as the key to distinguish them (Fig. 2 and S4).” (in line 321–338)
[Comment 7] Discussion. I recommend to remove the first half of the first paragraph in Discussion. It is not linked with the results and except that there were other hundreds of genera (species) where morphometry was used. Shortening of the text is possible also in the next paragraphs. The writing of the Discussion would be easier if the working hypotheses and aims would be defined more precisely.
[Our response 7]: Thank you for your suggestions. We have made major changes to the discussion to make the content and purpose of the study clearer and easier to understand.
“Discussion
The identification of species boundaries is very important for the development and utilization of species and biodiversity conservation [43]. However, there are extensive crosses among species of Populus [38, 44, 45], resulting in a large number of morphological variations, which makes it extremely difficult to identify species of the genus by morphological traits. With the rapid development of next-generation sequencing technology, integrated phylogeny has been widely used in the study of species [46-48]. It has relatively high objectivity and maneuverability to define species by comprehensive means.
For the definition of Ka and St populations, we provide three strong evidence to support that Ka and St are the same taxa, so we should cancel the taxonomic status of St and merge it into Ka. First of all, Ka and St are distributed on the plateau, the habitat is very similar, and the distribution range is narrow, only in Sichuan Province has a stable population (Fig. 1). Secondly, based on the comparison of the morphological traits of leaves and capsules, the morphological traits of leaves had little difference between the two taxa, so it was impossible to distinguish the two taxa (Fig. 2, S2 and S4). The same results were obtained in the common garden experiments (Fig. S3). In the morphological traits of the capsules, only two traits (the ratio of length to width of capsule valve and the capsule split number) had significant differences between the two taxa (Fig. 2 and S4). In addition, although the two taxa can be distinguished by discriminant analysis and systematic cluster analysis, the difference between them is only shown in the above two traits. Finally, based on the analysis of genome-wide resequencing data, the phylogenetic relationship between Ka and St is very close, and the source of genetic components is exactly the same, only a small difference in the proportion of genetic components (Fig. 3). Moreover, the Fst between them is extremely small (Fig. 4D), indicating that they are the same taxa. Therefore, morphological variation within a certain range can not be used as an index to define a species, but should be explored comprehensively through the geographical distribution, morphological traits and molecular genetic data of a species in order to get objective and accurate results.
Hybridization is an important process in biological evolution, which is ubiquitous in nature and creates a complex evolutionary network for the tree of life [49-51]. Hybridization exists in 16–34% families and 6–16% genus of plants in nature [52], which can give plants richer variation, stronger adaptability and competitiveness, and make them dominant in community regeneration [53, 54]. Firstly, based on the morphological traits and geographical distribution, we inferred that Ka and St are hybrids, and Si is one of its parents. Both of them have the typical traits of Si, such as the capsule is 2-valve, long ovate, the sprouting branches and leaves with weak growth are inverted wedge, and the seedlings of them are very similar to Si (Fig. S2 and S3). With the advent of the genome era, many outstanding problems have been broken through, which can further reveal the process and mechanism of hybrid speciation while identifying natural hybridization. Therefore, we also carried out a variety of analyses (ABBA-BABA, HyDe, TreeMix, ABC model and IBD) based on the whole genome SNPs to prove that Ka and St originated from the hybridization of Xi and Si, and there was a strong gene flow from Xi to them (Fig. 4, 5 and 6B).
Furthermore, through PSMC analysis, it was found that the effective historical population size of parents and hybrids all had a contraction period (Fig. 6A), which was related to the drastic change of Quaternary glacial climate on the QTP, and this phenomenon existed in many organisms, such as walnut [55], ironwood tree [56] and birch [57] and panda [58]. However, because the populations of Ka and St were endowed heterosis, their population size rebounded rapidly during the glacial period (Fig. 6A), which indicated that they had strong persistence and resilience compared with their parents during the Quaternary glacial period. Moreover, we also found a large number of genes related to stress resistance and growth regulation in hybrids based on positive selection analysis (Fig. 7). It is possible that these genes make Ka and St stronger adaptability, and the suitable distribution area of the taxa may gradually expand in the future (Fig. 8D).” (in line 541–597)
[Comment 8] References. Unfortunately, the authors did not follow the Instructions for Authors, both what concerns the citation in text and also the list of references. All cited references in text are in the list of references. The references are ordered as they appeared in the text. It is necessary to replace the cited references by consecutive numbers and correct all formatting imperfections in the list of references. In the list of references there is one item doubled (Echenwalder, J. and Eckenwalder, J.) and after removal one of them the numbering in the list of references should be altered.
[Our response 8]: Thank you for your suggestion. We have revised the format of references in accordance with “Instructions for Authors” of “Forests”.
[Comment 9]
Entire manuscript replace ~ by N-dash (from–to)
Entire manuscript insert space after coma in citations
Ln 6–7 complete address
Ln 32 Salicaceae not in Italics
Ln 35 Eckenwalder. In the list of references there are two citations of the same author and book chapter (2 and 8), although with not identical family name. The citations are imperfectly cited. The correct citation is as
follows:
Eckenwalder, J.E. 1996. Systematics and evolution of Populus. In: Stettler, R.F., Bradshaw, H.D. Jr, Heilman, P.E., Hinkley, T.M., editors. Biology of Populus and its Implications for Management and Conservation. Part I, Chapter 1. NRC Research Press, National Research Council of Canada, Ottawa, Canada, pp. 7–32.
Ln 86 Republicae
Ln 87 Schneideri lower case „s” schneideri
Ln 104 Populus tomentosa twice (delete one) and in italics
Ln 124, 205, 380–382, 392, 468, 470–471, 521, 526, 585, 733 use N-dash and replace tilda by N-dash
Ln 128–129 including five P. kangdingensis and nine P. schneideri var. tibetica.
Ln 134–135 describe the data matrix in detail (14 populations × 5 trees × 50 leaves ?), how many trees were used for morphometry of leaves and capsules and/or for resequencing
Ln 138 the capsules of trees at most sample sites; remove had – it is redundant
Ln 146–147 insert space between “ and E or N, respectively
Ln 151 traits instead of index (also on other places in the text) e.g. quantitative traits
Ln 170, 348 common garden (common garden experiments)
Ln 172 describe the „slight modification” of CTAB
Ln 173–175 how many samples per site or individuals per species were sent for genome resequencing?
Ln 184, 194, 207–208, 213, 230–231, 252, 273–274, 282, 408–484, 492–497, 569, 571 variates in Italics
Ln 205, 210, 380–382, 553–554 N-dash (minus, from–to (e.g. pages) and inserted sentence)
Ln 230–232, 273, 512 Italics
Ln 291 from 3400 to 3600 m
Ln 299 five
Ln 379 “,” coma (there are two different symbols for coma)
Ln 392 without spaces
Ln 326, 416, 424, 433, 439–441, 456, 457, 490, 512, 514, 565, 609, 614, 624, 628, 633, 645, 650, 651, 652, 655, 657, 667, 681, 389, 705–715 var. not in Italics. Prior page 326 it is OK
Ln 398 which method of cluster analysis and clustering matrix were used
Ln 597 species or varieties?
Ln 691 remove underline
References
- Format does not correspond to the instructions for authors! The Editor compiled and published
Instructions for Authors to help them to prepare a manuscript with adequate formatting. Respect these recommendations.
- In the text, reference numbers should be placed in square brackets [ ], and placed before the
punctuation; for example [1], [1–3] or [1,3]. I have noticed the authors prepared the complete list of references in ascending order as they appear in manuscript, but it is necessary to replace them by numbers at revision.
- Journal names should be abbreviated and in capitals, e.g. American Journal of Botany = Amer. J. Bot., Science, Nature Communications = Nat. Commun., Journal of Sichuan Forestry Science and
Technology = J. Sichuan For. Sci. Technol., Nature Genetics = Nat. Genet., Molecular Biology and Evolution = Mol. Biol. Evol., Molecular Ecology Resources = Mol. Ecol. Resour. etc. There are many home pages with abbreviated journal names, use one of them, e.g. https://images.webofknowledge.com/images/help/WOS/A_abrvjt.html.
- Publication year should be on the proper place and in Bold and volume in Italics instead of Bold.
- Page range should be with N-dash, instead of a hyphen “-“, e.g. 352–358.
- Ln 735, 742, 744, 746, 750, 755, 760, 763, 765, 769, 771, 775, 780, 783, 814, 819, 827, 829, 834, 837 Latin names in Italics
- Ln 746, 763–764, 790–791, 792–793, 802, 812 – replace the first capital letters in paper titles published in journals by lowercase ones, except the geographic names and species names with geographic or personal meaning e.g. Norway spruce, Scots pine, Engelman spruce, etc., e.g. “Phylogenomics and biogeography of Populus based on comprehensive sampling reveal deep-level relationships and multiple intercontinental dispersals” (in contrast with book titles where the first letters should be written with capitals).
Fig. S1 Use the map with Latin names, not with Chinese characters
There are two figures Fig. S1. Should not they have other number or the first one is invalid?
In the second Fig. S1 use N-dash – (A–H) and (I–P); insert space after “var.“
Fig. S2 Use (A–I) (J–R); insert space after “var.“
Fig. S5 “var.” not in Italics
Fig. S6 (Dxy), in Italics and xy in subscript – Dxy
Fig. S8 “var.” not in Italics
Table S2Column Altitude – replace ~ by N-dash
Columns L, M and N – Valve
Note: Table S4Column Description
Start the description in all cells by a Capital letter
[Our response 9]: Thank you for your suggestions. Everything has been revised as suggested.
Thank you for your comments. All questions are answered in the text.

Reviewer 2 Report
Comments and Suggestions for Authors
The manuscript presented for review is focused on the classification and origin of Populus kangdingensis and Populus schneideri var. tibetica, and verification of the rationality of their classification and treatment on the base of the comprehensive study of morphological taxonomy, ecological geography and molecular systematics. P. kangdingensis and P. schneideri var. tibetica are two taxa of Sect. Tacamahaca, endemics to southwest China which are mainly distributed on the surface of Qinghai-Tibet Plateau from 3400 79 m to 4000 m above sea levelр in western Sichuan. They are also native poplar species, which are widely planted and play an important role in the plateau ecological construction, landscape construction, production of medium and small diameter wood and paid carbon wood. They also have excellent adaptability to alpine and arid habitat and important breeding value in poplar breeding on Qinghai-Tibet Plateau. The two taxa are very similar in morphology and habitat, and are in doubt in present taxonomy. In the present study, on the base of phylogenetic analysis by whole genome resequencing was showed that the two taxa were hybrid progenies of P. xiangchengensis and P. simonii. Through gene flow detection and genetic differentiation analysis, it was found that there was still strong gene flow from P. xiangchengensis to P. kangdingensis and P. schneideri var. tibetica, and there was almost no differentiation between the two taxa. Therefore, P. schneideri var. tibetica should be treated as a variety of P. kangdingensis. Also, on the base of PSMC and ABC models, the population history was reconstructed, and it was found that the two taxa belonged to hybrid origin, and the change of population size was closely related to the Quaternary ice age. In addition, was also concluded that the hybrid populations have better adaptability, due to the growth regulatory genes selectively differentiated among them. At the same time, the study provides a novel and comprehensive method for the classification of Populus.
The manuscript is constructed according to requirements of “Forests”. The research methods applied are appropriate, comprehensive and sufficient to achieve the objectives of the study. The illustrative material is representative and of good quality. The statistical analysis applied complement and support the results.
A continuation, some recommendations are given:
Abstract:
The opening sentence is very general and states a well-known fact. So, my suggestion for the abstract beginning is: “Poplar are various important woody plants of the genus Populus, Salicaceae. Due to their characteristics of fast growth, easy reproduction and strong adaptability, it is one of the most important afforestation tree species in the world, and plays an important role in global wood production and ecological environment construction. The extensive hybridization in the genus makes it taxonomy very confused, especially in the Sect. Tacamahaca….”
Introduction
Lines 39-41“In addition, poplars also have good experimental characteristics, such as dioecious plants, easy hybridization, high compatibility, rapid growth, asexual reproduction…” : - My suggestion for this sentence is: “In addition, poplars also have good experimental characteristics, such as dioecious plants, easy hybridization, high compatibility, rapid growth, asexual (vegetative) reproduction along with the sexual one…”
Lines 103-104: “At present, although there are some studies on the classification, phylogeny and breeding of Populus tomentosa and Populus tomentosa (Wan, et al. 2013, Chen, et al. 2007) …” - Populus tomentosa is repeated twice - to correct - either add the other species, or if none, delete the repetition, and also Populus tomentosa should be in "italics"
Furthermore, the entire passage “At present, although there are some studies on the classification, phylogeny and breeding of Populus tomentosa and Populus tomentosa (Wan, et al. 104 2013, Chen, et al. 2007), there is no special research on them. Accurate classification is the basis of all development, utilization and scientific research.” needs editing.: “The development, utilization and scientific research on these taxa require an accurate classification. At present, there is no special research on them, although there are some studies on the classification, phylogeny and breeding of Populus tomentosa and Populus tomentosa (Wan, et al. 104 2013, Chen, et al. 2007).”
Results
Figure 1 to be moved from „Results“ to „Material and Methods“, immediately after its citation, after line 131.
References:
The references are not properly cited in the text
According to “Instructions for Authors” of “Forests”: “In the text, reference numbers should be placed in square brackets [ ], and placed before the punctuation; for example [1], [1–3] or [1,3]….”,
In conclusion, this manuscript is recommended for publication in “Forests”, after consideration of the remarks shown.
Author Response
Dear Editor:
We would like to thank you and the Reviewers for your valuable comments and suggestions that helped us further our understanding in this field as well as improved our manuscript. After carefully considering the reviewers’ comments, we have revised our manuscript point-by-point. For the major comments, we had responded to each as listed below. For your convenience, all changes made in the manuscript were tracked.
Please do not hesitate to contact me in case the manuscript has any further issues.
Thank you very much for your help.
We are looking forward to hearing from you soon.
Yours sincerely,
Xue-Qin Wan, Ph.D.
Professor of forest tree genetics and breeding
College of Forestry,
Sichuan Agricultural University, Chengdou 611130, China
E-mail: [email protected]
-------------------------------------------------------------------------------------------------------
Responses to reviewer 2:
[Comment 1] Abstract: The opening sentence is very general and states a well-known fact. So, my suggestion for the abstract beginning is: “Poplar are various important woody plants of the genus Populus, Salicaceae. Due to their characteristics of fast growth, easy reproduction and strong adaptability, it is one of the most important afforestation tree species in the world, and plays an important role in global wood production and ecological environment construction. The extensive hybridization in the genus makes it taxonomy very confused, especially in the Sect. Tacamahaca….”
[Our response 1]: Thank you for your suggestion. We have revised the original statement according your suggestion.
The original statement is “Classification is the basis for human understanding, research, development...especially in the Sect. Tacamahaca.” , and we have revised it to “Poplar not only has important ecological and economic value, but also is a model woody plant in scientific research. However, due to the rich morphological variation and extensive interspecific hybridization, the taxonomy of the genus Populus is very confused, especially in the Sect. Tacamahaca.” (in line 10–13).
[Comment 2] Introduction Lines 39-41“In addition, poplars also have good experimental characteristics, such as dioecious plants, easy hybridization, high compatibility, rapid growth, asexual reproduction…” : - My suggestion for this sentence is: “In addition, poplars also have good experimental characteristics, such as dioecious plants, easy hybridization, high compatibility, rapid growth, asexual (vegetative) reproduction along with the sexual one…”
Lines 103-104: “At present, although there are some studies on the classification, phylogeny and breeding of Populus tomentosa and Populus tomentosa (Wan, et al. 2013, Chen, et al. 2007) …” - Populus tomentosa is repeated twice - to correct - either add the other species, or if none, delete the repetition, and also Populus tomentosa should be in "italics"
Furthermore, the entire passage “At present, although there are some studies on the classification, phylogeny and breeding of Populus tomentosa and Populus tomentosa (Wan, et al. 104 2013, Chen, et al. 2007), there is no special research on them. Accurate classification is the basis of all development, utilization and scientific research.” needs editing.: “The development, utilization and scientific research on these taxa require an accurate classification. At present, there is no special research on them, although there are some studies on the classification, phylogeny and breeding of Populus tomentosa(Wan, et al. 104 2013, Chen, et al. 2007).”
[Our response 2]: Thank you for your suggestions. We have revised the Introduction according to your suggestions and corrected some mistakes.
The original statement is “In addition, poplars also have good experimental characteristics, such as dioecious plants, easy hybridization, high compatibility, rapid growth, asexual reproduction…” , and we have revised it to “In addition, poplars also have good experimental characteristics, such as dioecious plants, easy hybridization, high compatibility, rapid growth, asexual (vegetative) reproduction along with the sexual one, small genome (about 450 Mb), which makes poplar an ideal object for genetic breeding research [4-7].” (in line 40–44).
The original statement is “At present, although there are some studies on the classification, phylogeny and breeding of Populus tomentosa and Populus tomentosa (Wan, et al. 104 2013, Chen, et al. 2007), there is no special research on them. Accurate classification is the basis of all development, utilization and scientific research.” , and we have revised it to “The development, utilization and scientific research on these taxa require an accurate classification. At present, there is no special research on them, although there are some studies on the classification, phylogeny and breeding of Populus tomentosa [11, 15]. ” (in line 82–85).
[Comment 3] Results Figure 1 to be moved from “Results” to “Material and Methods”, immediately after its citation, after line 131.
[Our response 3]: Thank you for your suggestion. We have moved the Figure 1 from “Results” to “Material and Methods”, and added a citation after the original line 131. (in line 132)
[Comment 4] References:The references are not properly cited in the text
According to “Instructions for Authors” of “Forests”: “In the text, reference numbers should be placed in square brackets [ ], and placed before the punctuation; for example [1], [1–3] or [1,3]….”,
[Our response 4]: Thank you for your suggestion. We have revised the format of references in accordance with “Instructions for Authors” of “Forests”.
Thank you for your comments. All questions are answered in the text.

Round 2
Reviewer 1 Report
Ln 183 (1)
Ln 184 (2)
Ln 189–190 replace calculations by replications – The bootstraps values were obtained after 1000 replications
Ln 236 (pi)
Ln 247 minus should be as N-dash
Ln 256 minus should be as N-dash
Ln 397 minus should be as N-dash
Ln 430 from–to should be as N-dash
After the last author in the list of References does not end with coma, but only with dot. E.g. Hamzeh, M.; Dayanandan, S. Phylogeny ….
Ln 652 Book title – first letter in Capitals e.g. Poplars and Willows: Trees for Society and the Environment
Ln 648 Populus in Italics
Ln 648–649 Book title in Italics Biology of Populus and its Implications for Management and
Conservation
Ln 650, 653, 665 and 770 years in Bold
Ln 661 trichocarpa in Italics
Ln 664–665 Flora of China in Italics
Ln 665 Missouri
Ln 667 intermedia in Italics
Ln 661 trichocarpa in Italics
Ln 680 poplar
Ln 684–685 Doyle, J. J.; Doyle, J. L. A rapid DNA isolation procedure for small quantities of fresh leaf tissue. Phytochem. Bull. 1987, 19, 11–15.
Ln 704 lowercase “I”
Ln 765 Z, Y, M and Q in Capitals
Ln 769 Natural Hybridization and Evolution in Italics

Author Response
Dear Editor:
We would like to thank you and the Reviewers for your valuable comments and suggestions that helped us further our understanding in this field as well as improved our manuscript. After carefully considering the reviewers’ comments, we have revised our manuscript point-by-point. For the major comments, we had responded to each as listed below. For your convenience, all changes made in the manuscript were tracked.
Please do not hesitate to contact me in case the manuscript has any further issues.
Thank you very much for your help.
We are looking forward to hearing from you soon.
Yours sincerely,
Xue-Qin Wan, Ph.D.
Professor of forest tree genetics and breeding
College of Forestry,
Sichuan Agricultural University, Chengdou 611130, China
E-mail: [email protected]
-------------------------------------------------------------------------------------------------------
Responses to reviewer 1:
Comments – 2
Ln 183 (1)
Ln 184 (2)
Ln 189–190 replace calculations by replications – The bootstraps values were obtained after 1000 replications
Ln 236 (pi)
Ln 247 minus should be as N-dash
Ln 256 minus should be as N-dash
Ln 397 minus should be as N-dash
Ln 430 from–to should be as N-dash
After the last author in the list of References does not end with coma, but only with dot. E.g. Hamzeh, M.; Dayanandan, S. Phylogeny ….
Ln 652 Book title – first letter in Capitals e.g. Poplars and Willows: Trees for Society and the Environment
Ln 648 Populus in Italics
Ln 648–649 Book title in Italics Biology of Populus and its Implications for Management and
Conservation
Ln 650, 653, 665 and 770 years in Bold
Ln 661 trichocarpa in Italics
Ln 664–665 Flora of China
Ln 665 Missouri
Ln 667 intermedia in Italics
Ln 680 poplar
Ln 684–685 Doyle, J. J.; Doyle, J. L. A rapid DNA isolation procedure for small quantities of fresh leaf tissue. Phytochem. Bull. 1987, 19, 11–15.
Ln 704 lowercase “I”
Ln 765 Z, Y, M and Q in Capitals
Ln 769 Natural Hybridization and Evolution
[Our response 1]: Thank you for your suggestions. We have carefully revised the above problems as required, and the changes are highlighted in red. In addition, with regard to the question of changing Ln 247, 256, 397 and 430 to N-dash, we were unable to distinguish the difference in appearance, so we directly copied the symbols suggested by the reviewer and pasted them into the article.
